# The actin-modulating protein synaptopodin mediates long-term survival of dendritic spines

Kenrick Yap[1], Alexander Drakew[1], Dinko Smilovic[1,2], Michael Rietsche[1], Mandy H Paul[1], Mario Vuksic[1,2], Domenico Del Turco[1], Thomas Deller[1]*

[1]Institute of Clinical Neuroanatomy, Dr. Senckenberg Anatomy, Neuroscience Center, Goethe University Frankfurt, Frankfurt, Germany; [2]Croatian Institute for Brain Research, School of Medicine, University of Zagreb, Zagreb, Croatia

**Abstract** Large spines are stable and important for memory trace formation. The majority of large spines also contains synaptopodin (SP), an actin-modulating and plasticity-related protein. Since SP stabilizes F-actin, we speculated that the presence of SP within large spines could explain their long lifetime. Indeed, using 2-photon time-lapse imaging of SP-transgenic granule cells in mouse organotypic tissue cultures we found that spines containing SP survived considerably longer than spines of equal size without SP. Of note, SP-positive (SP+) spines that underwent pruning first lost SP before disappearing. Whereas the survival time courses of SP+ spines followed conditional two-stage decay functions, SP-negative (SP-) spines and all spines of SP-deficient animals showed single-phase exponential decays. This was also the case following afferent denervation. These results implicate SP as a major regulator of long-term spine stability: SP clusters stabilize spines, and the presence of SP indicates spines of high stability.

*For correspondence:
t.deller@em.uni-frankfurt.de

Competing interests: The authors declare that no competing interests exist.

## Introduction

Dendritic spines are protrusions found on the majority of excitatory neurons in vertebrate brains. They form characteristic axo-spinous synapses and play an important role in integrating afferent synaptic activity with postsynaptic activity (*Yuste and Denk, 1995*). The geometry of a spine, in particular, the length of the spine neck and the size of the spine head are considered structural correlates of synapse function: While the length of the spine neck has been linked to the biochemical and electrical isolation of the spine compartment (*Yuste et al., 2000*; *Yuste, 2013*), the size of the spine head has been positively correlated with AMPA-receptor density, synaptic strength and spine stability (*Matsuzaki et al., 2004*; *Kasai et al., 2010*; *McKinney, 2010*). At the behavioral level, a critical role of spines in learning, memory formation and cognition has been discussed (*Segal, 2005*; *Bourne and Harris, 2007*; *Kasai et al., 2010*; *McKinney, 2010*; *Roberts et al., 2010*) and, indeed, a recent study has shown that spines are both necessary and sufficient for memory storage and memory trace formation (*Abdou et al., 2018*).

The function of spines does not only depend on the shape and size of their outer membranes but also on the molecular machinery within the spine compartment. A cellular organelle unique to spines is the spine apparatus (*Gray, 1959*; *Spacek, 1985*), which consists of stacked endoplasmic reticulum (ER) and which modifies spine $Ca^{2+}$ transients (*Korkotian and Segal, 2011*) and second messenger dynamics (*Cugno et al., 2018*). Synaptopodin (SP; GenBank accession number: NM_001109975.1) is an essential component of the spine apparatus and is required for its formation. It is an actin-modulating protein found in a 100 kDa isoform in telencephalic neurons and a 110 kDa isoform in kidney podocytes (*Mundel et al., 1997*; *Asanuma et al., 2005*). SP is almost linear because of its high proline content and contains several phosphorylation sites and two PPXY motifs for protein-protein

interactions (*Mundel et al., 1997*; *Asanuma et al., 2005*; *Asanuma et al., 2006*; *Faul et al., 2007*). Expression of SP in mice starts postnatally (*Mundel et al., 1997*; *Czarnecki et al., 2005*), increases with maturation (*Czarnecki et al., 2005*) and declines with age (*Sidhu et al., 2016*). SP-deficient mice show deficits in synaptic plasticity (*Deller et al., 2003*; *Jedlicka et al., 2009*; *Vlachos et al., 2009*; *Vlachos et al., 2013b*; *Zhang et al., 2013*; *Korkotian et al., 2014*; *Jedlicka and Deller, 2017*) and adult SP-deficient mice show impaired spatial learning (*Deller et al., 2003*).

The function of SP in spines is not limited to the formation of spine apparatus organelles. SP also affects the actin spinoskeleton either directly by stabilizing F-actin (*Mundel et al., 1997*; *Okubo-Suzuki et al., 2008*) or indirectly via binding to alpha-actinin-2, Cdc42, RhoA, or myosin V (*Asanuma et al., 2005*; *Kremerskothen et al., 2005*; *Asanuma et al., 2006*; *Faul et al., 2007*; *Yanagida-Asanuma et al., 2007*; *Jedlicka and Deller, 2017*; *Konietzny et al., 2019*). It is likely that SP, via its connection to the actin cytoskeleton, is also connected to the post-synaptic density (PSD), which plays an important role in synaptic stabilization (*El-Husseini et al., 2000*; *Ehrlich et al., 2007*; *Yoshihara et al., 2009*; *Meyer et al., 2014*). Accordingly, SP has been suggested to influence the geometry and stability of spines (*Deller et al., 2000a*). Indeed, short-term imaging experiments performed in dissociated cultures (*Okubo-Suzuki et al., 2008*; *Vlachos et al., 2009*; *Konietzny et al., 2019*) and acute slices (*Zhang et al., 2013*) implicated SP in plasticity-induced spine head expansion. We have now expanded on these previous studies, which focused on the short-term effects of SP on spine geometry, and have used 2-photon time-lapse imaging of identified SP+ and SP− granule cell spines to address the question of whether SP is important for long-term spine stability.

## Results

### Synaptopodin is present in large granule cell spines in vivo

Previous work reported a positive correlation between the plasticity-related protein SP and spine head size (*Okubo-Suzuki et al., 2008*; *Vlachos et al., 2009*; *Zhang et al., 2013*). Since SP-deficient mice show deficits in synaptic plasticity (*Deller et al., 2003*; *Jedlicka et al., 2009*; *Zhang et al., 2013*; *Grigoryan and Segal, 2016*), we speculated that SP-deficient neurons might also have smaller spine head sizes, providing a structural explanation for the plasticity phenotype. Accordingly, we analyzed and compared spine head size in Alexa568-filled wildtype and mutant granule cells (*Figure 1A,B*) but could neither detect differences in average spine head sizes (*Figure 1C*) nor in spine head size distributions (*Figure 1D*). Next, we re-visited the relationship between SP and spine head size that had been suggested earlier (*Okubo-Suzuki et al., 2008*; *Vlachos et al., 2009*; *Zhang et al., 2013*). Using Alexa 568-injected granule cells from eGFP-SP-tg mouse brain (*Figure 1E*; *Vlachos et al., 2013a*), we first studied the fraction of SP-positive spines and found SP clusters in ~14% of spines (*Figure 1F*). Spines containing SP (SP+) were significantly larger than spines without SP (SP-) (*Figure 1G*) and showed a right-shifted cumulative frequency distribution (*Figure 1H*). In addition, SP cluster size was positively correlated with spine head size (*Figure 1I*), demonstrating that large SP clusters are typically found in large spines. We conclude from these observations, that SP is indeed tightly correlated with spine size. However, lack of SP did not affect average spine head size, suggesting that SP is either not a major regulator of spine head size or that its loss is compensated for in the SP-deficient mouse.

This finding raised the question of what is the role of SP in large spines? Since SP is an actin-modulating plasticity-related protein, it has been speculated that SP could also regulate spine stability (e.g. *Deller et al., 2000a*). As previous reports have shown that large cortical spines are more stable than small spines (*Kasai et al., 2003*; *Matsuzaki et al., 2004*; *Bourne and Harris, 2008*; *Kasai et al., 2010*; *McKinney, 2010*), we speculated that the presence of SP in these spines could explain or at least contribute to their higher stability. To test this hypothesis, we used an organotypic tissue culture preparation, which allowed us to follow not only spine geometry – which can also be done in vivo – but also the dynamics of SP clusters within spines over time.

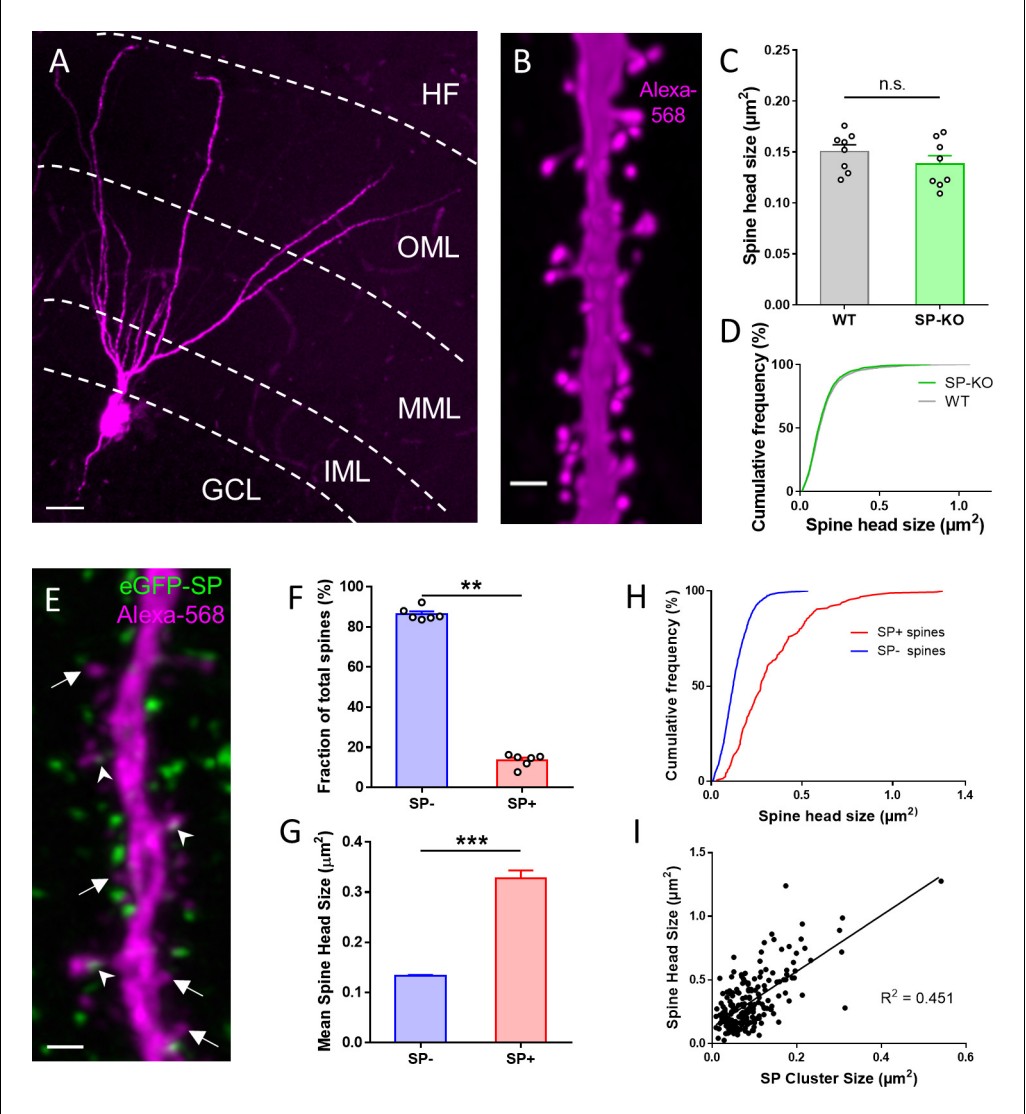

**Figure 1.** SP is associated with large granule cell spines in adult mouse dentate gyrus. (**A**) Granule cell located in the suprapyramidal blade of the dentate gyrus of a SP-knock-out (SP-KO) mouse intracellularly filled with the fluorescent dye Alexa568 (magenta; fixed tissue). Dendritic segments in the outer molecular layer (OML) were used for analysis. MML, middle molecular layer; IML, inner molecular layer; GCL, granule cell layer; HF, hippocampal fissure. Scale bar = 20 µm; Maximum intensity projection of a deconvolved z-stack. (**B**) Dendritic segment of a SP-KO granule cell shown at higher magnification. Scale bar = 1 µm; Maximum intensity projection of a deconvolved z-stack. (**C**) Spine head sizes of wildtype (WT) and SP-KO mice. n.s., not significant; p = 0.291, Mann–Whitney U-test. WT mice and SP-KO mice, n = 8 per group (three dendritic segments per animal; 1885 WT spines, 2158 SP-KO spines). (**D**) Cumulative frequency plots of spine head sizes of wildtype (grey) and SP-KO (green) mice. (**E**) Dendritic segment of a granule cell located in the suprapyramidal blade of the dentate gyrus of a Thy1-eGFP-SP-transgenic mouse bred on a SP-KO background (eGFP-SP-tg mouse) intracellularly filled with Alexa568 (magenta; fixed tissue; OML). Arrowheads point to eGFP-SP clusters (green) in SP-positive (SP+) dendritic spines. Arrows mark SP-negative (SP-) spines. Scale bar = 1 µm; Maximum intensity projection of a deconvolved z-stack. (**F**) Fractions of SP− (~86.5%) and SP+ (~13.5%) spines. **p<0.0022, Mann–Whitney U-test. eGFP-SP-tg mice, n = 6. (**G**) Mean spine head size of SP− (~0.133 µm$^2$) and SP+ (~0.328 µm$^2$) spines. ***p<0.0001, Mann–Whitney U-test. SP+ spines n = 200; SP− spines n = 1497. (**H**) Cumulative frequency plots of spine head sizes of SP− (blue) and SP+ (red) spines. SP+ spines n = 200; SP− spines n = 1497. (**I**) Correlation analysis of spine head size and SP cluster size: Spearman coefficient of correlation = 0.536, 95% C.I.: 0.426–0.630. ***p<0.0001. Linear regression analysis: R2 = 0.451. n = 200 spines.

## Synaptopodin is present in large granule cell spines in organotypic tissue cultures

Organotypic tissue cultures of the entorhinal cortex and hippocampus were generated from eGFP-SP-tg mice. In these cultures SP clusters were abundant in the molecular layer of the dentate gyrus (*Figure 2A*), similar to what has been described for wildtype animals after SP immunolabeling (*Deller et al., 2000b*; *Bas Orth et al., 2005*). To visualize single granule cells and their spines, tissue cultures were virally transduced on 2–3 days in vitro (DIV) using an AAV2-tdTomato virus. After approximately three weeks in vitro, tdTomato expressing granule cells were imaged and spines with SP (SP+) and without SP (SP-) were identified (*Figure 2B*). SP+ spines also contained SP-positive spine apparatus organelles (*Figure 2C*). SP-mRNA expression levels in granule cells of eGFP-SP-tg cultures were similar to SP-mRNA expression levels in granule cells of SP wildtype cultures (*Figure 2D*). This also held true for virus-injected eGFP-SP-tg cultures compared to virus-injected SP wildtype cultures (*Figure 2—figure supplement 1*). Fluorescence in situ-hybridization (FISH) for SP revealed that essentially all granule cells of eGFP-SP-tg cultures express the transgene (*Figure 2—figure supplement 1*). In line with the results obtained in vivo (*Figure 1C,D*), average spine head size and spine head size distribution were not significantly different between these and SP-deficient cultures (*Figure 2E,F*).

Next, we analyzed the fraction of SP+ and SP− spines and found that approximately 8% of all granule cell spines are SP+ (*Figure 2G*) and ~6.4% of all granule cell spines contained a spine apparatus organelle (not shown). Analysis of head sizes of SP+ and SP− spines (*Figure 2*; *Figure 2—figure supplement 2*) revealed that SP+ spines are on average much larger than SP- spines (SP- spines: $0.364 \pm 0.005\ \mu m^2$; SP+ spines: $0.801 \pm 0.029\ \mu m^2$; *Figure 2H*), similar to what we had observed in vivo (c.f. *Figure 1*). Likewise, cumulative frequency diagrams showed that the population of SP+ spines is right-shifted toward larger spine head sizes compared to SP- spines (*Figure 2I*). Finally, correlation analysis of SP cluster size and spine head size demonstrated a positive correlation between the two parameters (*Figure 2J*) as well as between spine apparatus area and spine head size (*Figure 2—figure supplement 3*). We conclude from these observations that regarding SP expression and distribution, granule cells in organotypic tissue cultures are highly similar to granule cells in vivo (c.f. *Figure 1*). Furthermore, we conclude that SP is preferentially found in large granule cell spines.

## Presence and size of synaptopodin clusters in vitro are tightly associated with bidirectional changes in spine head size

After demonstrating significant differences in average spine head size between SP+ and SP- spines, we wondered whether this relationship between SP and spine head size is also seen under dynamic conditions, that is in spines undergoing changes in their head size. For this, we studied how insertion and loss of SP clusters from spines affects spine head geometry (*Figure 3*). Single spines containing SP and spines devoid of SP were identified and spine head sizes were measured on two consecutive days (*Figure 3*). Four groups of spines were distinguished: Spines gaining SP (*Figure 3A*), spines losing SP (*Figure 3B*), spines maintaining SP (*Figure 3C*), and spines remaining SP- (*Figure 3D*). This analysis revealed a robust correlation between increases in spine head size and insertion of SP and decreases of spine head size and loss of SP from spines.

## The presence of synaptopodin in spines is associated with long-term spine survival

Next, we studied whether the presence of SP in a spine is associated with its long-term survival. This appeared to be a possibility, since SP protects actin filaments from disruption (*Okubo-Suzuki et al., 2008*) and – together with alpha-actinin-2 – can cause the formation of elongated and highly stable actin filaments (*Asanuma et al., 2005*).

To address this question, we used time-lapse imaging and followed single spines for two weeks (*Figure 4A*). During this time period, we determined their SP content, head sizes and fate (*Figure 4A,B*). This analysis revealed major differences in the survival time of SP+ and SP- spines (*Figure 4C–E*): Whereas the median survival time of SP+ spines was ~17.5 days, the median survival time of SP- spines was only ~6.8 days (*Figure 4C*).

In addition to this notable difference in median survival time, the curves that fitted our data turned out to be fundamentally different: Whereas SP- spines followed a single-phase exponential

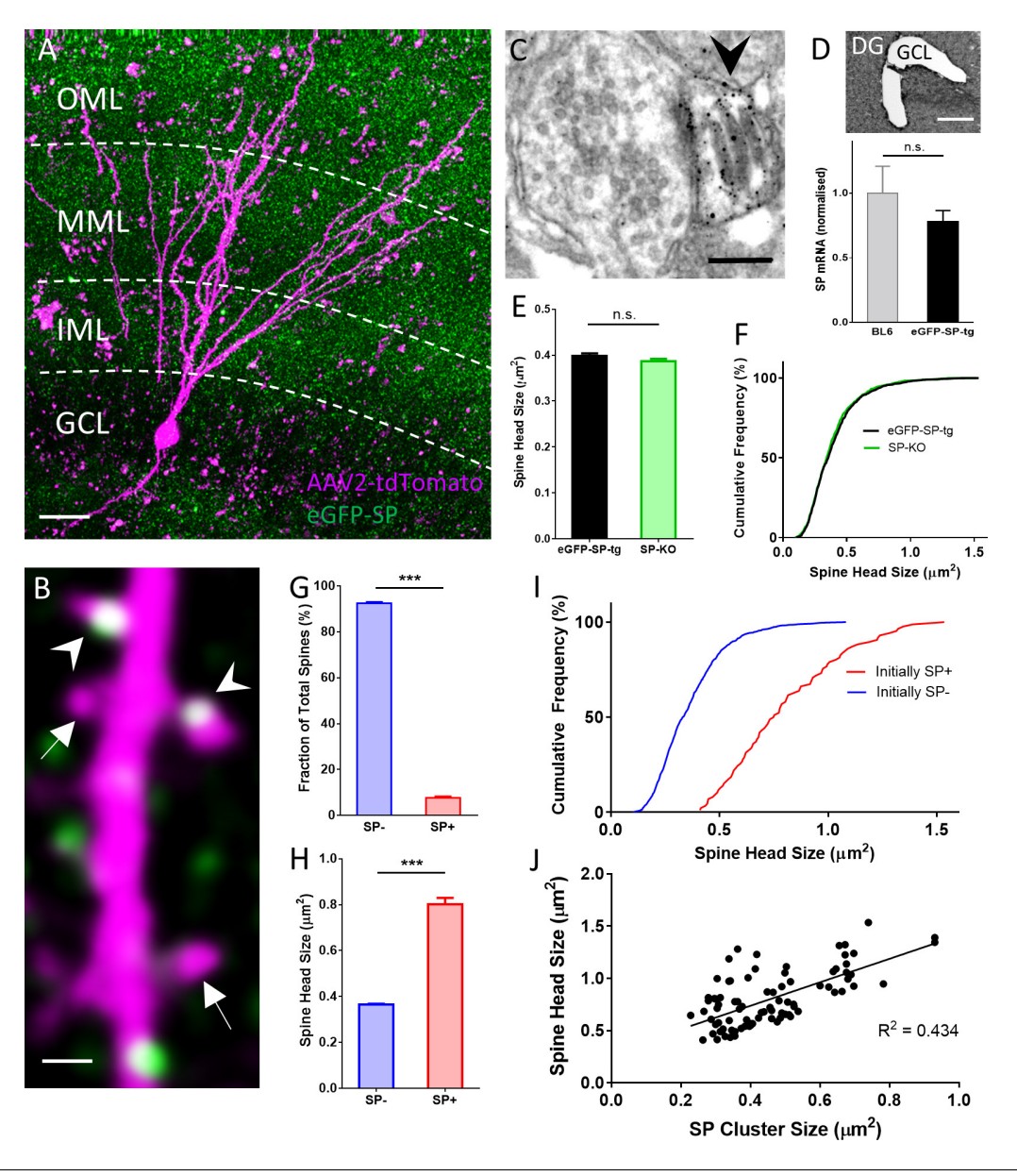

**Figure 2.** SP in dendritic spines of granule cells in organotypic tissue cultures of hippocampus. (**A**) Granule cells in organotypic entorhino-hippocampal tissue cultures (OTCs) of eGFP-SP-tg (green) mice were virally transduced (AAV2) with tdTomato (magenta) at day in vitro (DIV) 2–3. Dendritic segments located in the middle molecular layer (MML) or in the outer molecular layer (OML) were imaged. GCL, granule cell layer; IML, inner molecular layer. Scale bar = 20 μm; Maximum intensity projection of a deconvolved z-stack. (**B**) Single plane 2-photon image of a granule cell dendrite in the OML. Arrowheads point to SP+ spines; arrows indicate SP– spines. Scale bar = 1 μm. (**C**) Electron micrograph of a SP+ spine (arrowhead) containing an immunolabeled spine apparatus in an OTC from an eGFP-SP-tg mouse. Scale bar = 0.2 μm. (**D**) SP-mRNA levels in micro-dissected granule cell layers (GCL; upper panel) from OTCs of eGFP-SP-tg and C57BL/6J (BL6) wildtype mice were not significantly different (lower panel). p=0.353, Mann–Whitney U-test. Number of BL6 tissue cultures n = 6; number of eGFP-SP-tg tissue cultures n = 10. DG, dentate gyrus. Scale bar = 100 μm. (**E**) Mean spine head sizes of eGFP-SP-tg (black) and SP-KO (green) granule cells. n.s., not significant. p=0.211, Mann–Whitney U-test. Number of spines: eGFP-SP-tg n = 1107; SP-KO n = 1113, obtained from 24 segments, one segment per culture. (**F**) Cumulative frequency plots showing the distribution of spine head sizes of these spines. (**G**) SP+ spines comprised 7.6% of the total spine population. ***p<0.0001, Mann–Whitney U-test. Percentage per eGFP-SP-tg dendritic segment; n = 24 segments. (**H**) SP+ spines were significantly larger than SP– spines. ***p<0.0001, Mann–Whitney U-test. Number of SP+ spines

*Figure 2 continued on next page*

*Figure 2 continued*

n = 87, SP− spines n = 1021. (I) Cumulative frequency plots showing the distribution of spine head sizes of SP+ and SP− spines. (J) SP cluster size and spine head size of SP+ spines are tightly correlated: Spearman coefficient of correlation = 0.565, 95% C.I.: 0.396–0.697. ***p<0.0001. Linear regression analysis: R2 = 0.434. n = 87 spines.
The online version of this article includes the following figure supplement(s) for figure 2:

**Figure supplement 1.** Fluorescence in-situ hybridization and laser microdissection/quantitative PCR for SP-mRNA.
**Figure supplement 2.** Measurement of dendritic spine head size and SP cluster size.
**Figure supplement 3.** Spine head size and spine apparatus size tightly correlate.

decay function, the decay of SP+ spines could only be fitted using a conditional two-stage exponential decay function (see below; *Figure 4C*). Based on these two functions the distributions of long-term survival times (*Figure 4D,E*) of the SP- and SP+ spine populations were calculated (see Materials and methods). Finally, a data-driven simulation of spine survival was performed to test whether the survival times of the two spine populations are different. This approach revealed – within the parameters used – that SP- and SP+ spines differ significantly in their survival times (*Figure 4F*).

Since SP+ spines are larger than SP- spines (*Figures 1G* and *2H*), the observed effect on spine survival could simply be the result of differences in head size between the two groups. As large spines have been shown to be more stable (*Kasai et al., 2003*; *Matsuzaki et al., 2004*; *Bourne and Harris, 2008*; *Kasai et al., 2010*; *McKinney, 2010*), this would be a straightforward explanation. We controlled for this possibility by comparing the survival times of SP+ and SP- spines of equal head size (matched pairs; *Figure 4G*). Even under these highly constrained conditions, the difference in spine survival between SP+ and SP- spines was still observed (*Figure 4H*). In line with this, the overall distributions (*Figure 4I*), the relative frequency distributions of spine survival times of the calculated total populations of spines (*Figure 4J*), as well as simulations of spine survival (*Figure 4K*) were comparable to those found for all spines, that is non-size-matched spines. Finally, we tested whether the time point at which SP had entered a spine plays a role. For this we compared 'timed' SP+ spines, which had just become SP+ (between day −1 and 0) with SP- spines that had stayed SP- (between day −1 and 0). The survival curve of these 'timed' SP+ spines (*Figure 4—figure supplement 1*) was again highly similar to the curve of the total population of SP+ spines (*Figure 4C*).

## Synaptopodin is correlated with the long-term survival of spines irrespective of spine head size

Since the stability of large spines is higher than the stability of small spines (*Kasai et al., 2003*; *Matsuzaki et al., 2004*; *Bourne and Harris, 2008*; *Kasai et al., 2010*; *McKinney, 2010*), we wondered whether this difference could be linked to the presence of SP within large spines. This appeared to be a possibility since SP is tightly associated with the population of large and stable spines and rare in the group of small and less stable spines (*Figure 5A,B*). Should the increased stability of large spines depend on SP we made two testable predictions: (1) If SP is present in small spines, it should increase survival of these spines irrespective of their size, and, (2) if SP is absent from large spines, these spines should be pruned faster in spite of their large size. To test these predictions, we subdivided SP+ and SP- spines into four size classes (*Figure 5B*; very small, small, medium and large spines) and compared the survival times of SP- and SP+ spines belonging to each of these classes. To test the first hypothesis, we analyzed the three smaller size groups, which comprise the majority (~95%) of all spines (*Figure 6A*). SP was completely absent from the very small spine group, in which ~ 41% of all SP- spines are found (*Figure 5B*). This spine population rapidly decayed with time following a single exponential decay function (*Figure 5C–F*).

In the small spine group ~47.8% of all SP- and ~17.4% of all SP+ spines are found (*Figure 5B*), with SP+ spines representing ~5% of all spines in this group. Compared to the SP- spines, which followed a single exponential decay curve, SP+ spines showed a longer survival time and followed a conditional two-stage exponential decay function (*Figure 5G–J*). In the group of medium-sized spines ~ 9.5% of SP- and ~39.5% of SP+ spines are found, with SP+ spines representing ~60% of the spines in this size group. Again, SP- spines followed a single exponential decay curve, whereas SP+ spines showed longer survival times and their decay was compatible with a conditional two-stage

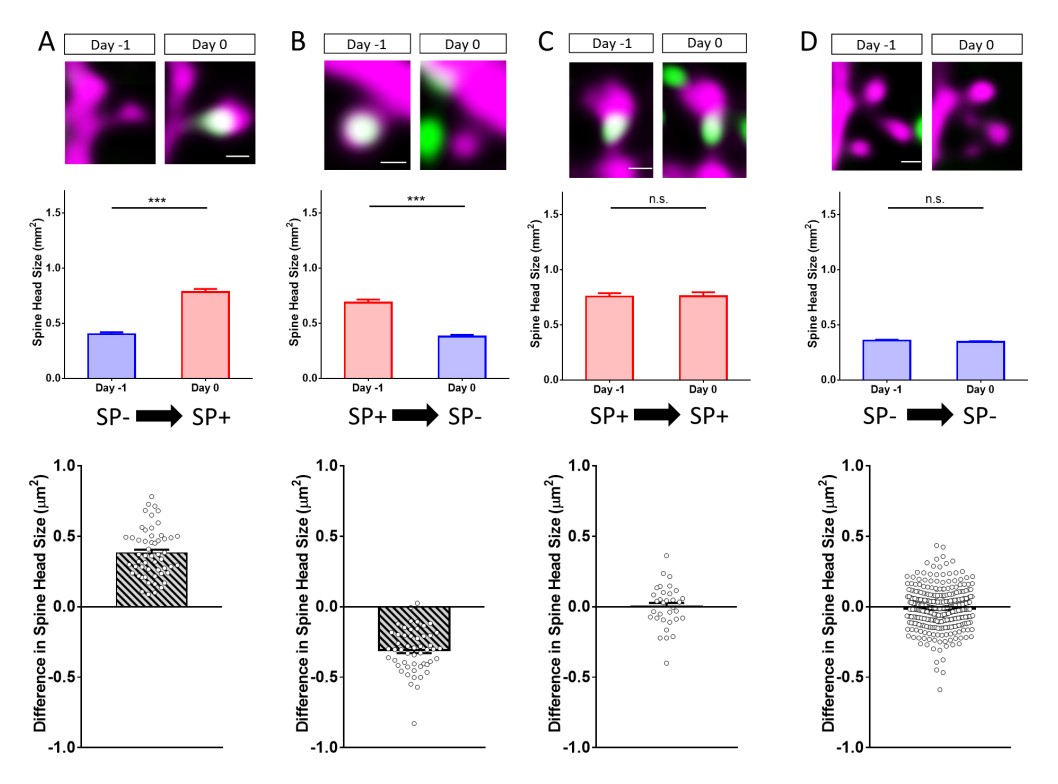

**Figure 3.** Changes in spine SP content are associated with bidirectional changes in spine head size. Sample spines (upper panels), mean spine head size changes (middle panels) and differences in spine head size of individual spines (lower panels) are illustrated. (**A**) Example of a spine that became SP+ (green or white co-localized with magenta spine) between day −1 and 0. The mean maximum cross-sectional spine head area of spines of this group increased significantly. ***p<0.0001, Wilcoxon matched-pairs signed rank test. Number of spines n = 54. Scale bar = 0.5 µm. (**B**) Example of a spine that was SP+ and became SP- between day −1 and 0. The mean maximum cross-sectional spine head area of spines of this group decreased significantly. ***p<0.0001, Wilcoxon matched-pairs signed rank test. Number of spines n = 52. Scale bar = 0.5 µm. (**C**) Example of a spine that remained SP+. The mean cross-sectional spine head area of spines of this group did not change significantly. p=0.850, Wilcoxon matched-pairs signed rank test. Number of spines n = 33. Scale bar = 0.5 µm. (**D**) Example of a spine that remained SP-. The mean cross-sectional spine head area of spines of this group did not change significantly. p=0.060, Wilcoxon matched-pairs signed rank test. Number of spines n = 340. Scale bar = 0.5 µm; Single plane deconvolved 2-photon images.

exponential decay function (*Figure 5K–N*). We conclude from these data that small and medium-sized spines survive longer and follow a conditional two-stage decay kinetic if SP is present.

Finally, we looked at the population of large spines. Although these spines only account for ~4.9% of all spines, they are considered especially important in the context of memory storage (*Segal, 2005*; *Bourne and Harris, 2007*; *Kasai et al., 2010*; *McKinney, 2010*; *Abdou et al., 2018*). In this group of spines only ~1.7% of the SP- spines are found whereas ~43% of SP+ spines are part of this subgroup. SP+ spines represent ~65% of the spines in this size group. Comparing spine pairs of equal head sizes (*Figure 5O*) major differences in their survival and decay kinetics were revealed: Whereas SP-spines followed a single exponential function, SP+ spines were best modeled using the conditional two-stage decay function (*Figure 5P–S*). We conclude that large SP- spines are pruned fast. Their survival is comparable to that of very small (*Figure 5C*), small (*Figure 5G*) and medium-sized (*Figure 5K*) spines, suggesting that it is the presence of SP within spines rather than spine size per se which determines long-term spine survival.

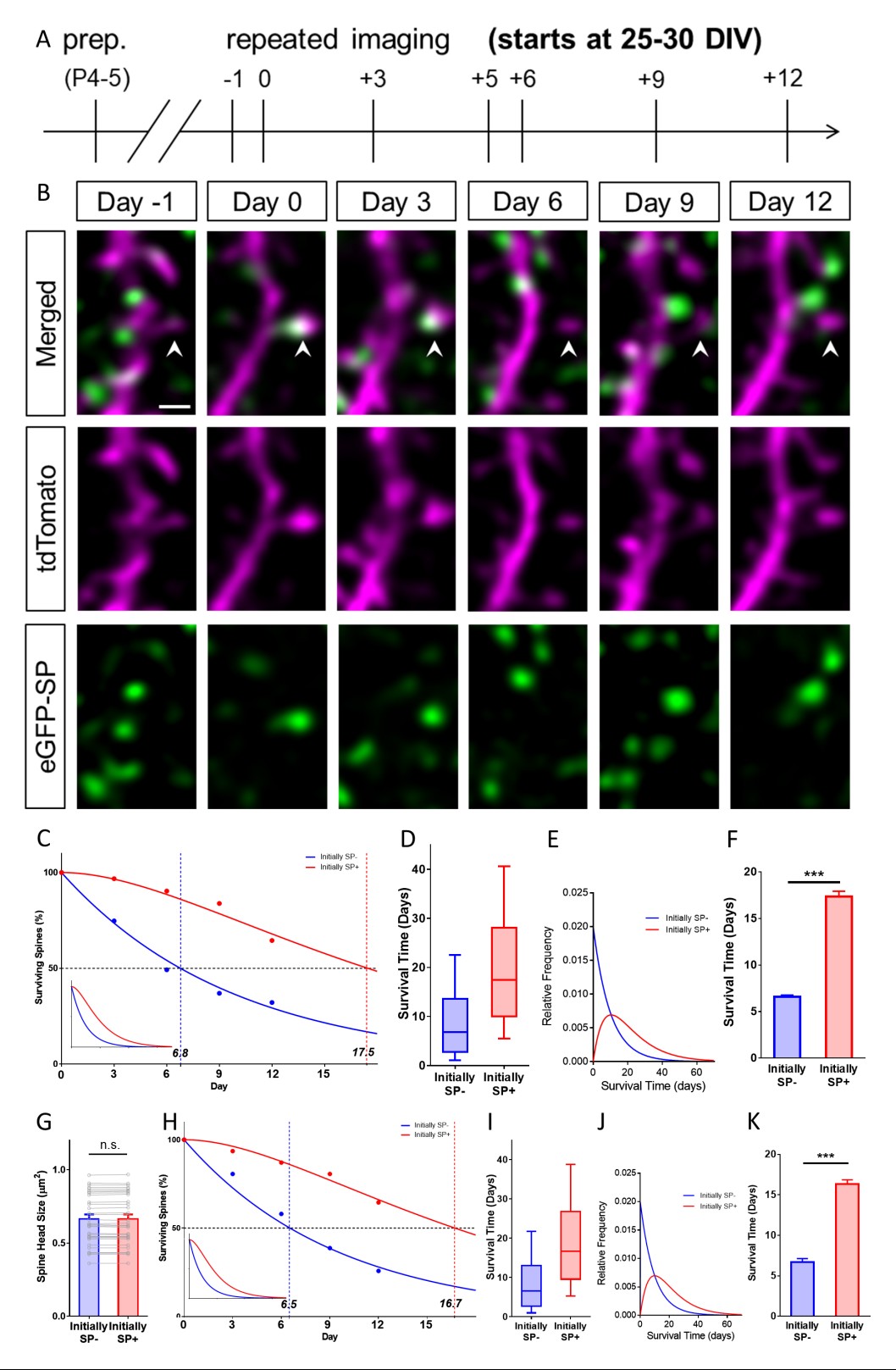

**Figure 4.** SP+ spines are highly stable spines. (**A**) Schematic of the experimental design. OTCs were prepared at postnatal day 4–5 from eGFP-SP-tg mice and allowed to mature for 25–30 DIV. Time-lapse 2-photon imaging of identified dendritic segments was performed at time points as indicated. (**B**) Time-lapse imaging of a granule cell dendrite over 14 days illustrates the dynamics of SP clusters within individual spines. Arrowhead points to a spine

*Figure 4 continued on next page*

*Figure 4 continued*

that became SP+ on day 0, stayed positive on day 3, lost SP between days 3 and 6 and then remained SP- until day 12. Scale bar = 1 µm; Single plane deconvolved 2-photon images. (**C**) The survival of SP+ (red curve) and SP-spines (blue curve) was studied from day 0 until day 12. Dots: observed fractions; curves: fitted to observation points. The median survival of initially SP- spines was ≈ 6.8 days whereas the median survival of initially SP+ spines was ≈ 17.5 days. Number of spines at day 0: SP+ = 31, SP- = 392. Inset: Calculated decay curves until day 70. SP-spines followed a single-phase decay function whereas SP+ spines followed a conditional two-stage decay function. (**D**) Survival times of the total populations of SP+ and SP- spines derived from the fitted survival curves. Boxes indicate median, lower and upper quartiles. Whiskers indicate 10th and 90th percentiles, respectively. (**E**) Relative frequency distributions of these survival times. (**F**) The survival time of SP+ spines was significantly longer than of SP- spines; computational model; ***p<0.0001, Mann–Whitney U-test; based on 10 samples of 31 SP+ and 392 SP- spines. (**G**) To control for differences in spine head size, SP+ and SP- spines of equal head size were matched. The spine head size of these pairs was not significantly different. p=0.310, Wilcoxon matched-pairs signed rank test. Number of size-matched pairs n = 31. (**H**) The decay curve of these size-matched spines was very similar to the decay curve of all SP+ and SP- spines. The median survival of SP- spines of this population was ≈ 6.5 days, whereas size-matched SP+ spines showed a median survival of ≈ 16.7 days. Inset: Calculated decay curves until day 70. (**I**) Survival times of the total populations of SP+ and SP- spines derived from the fitted survival curves. (**J**) Relative frequency distributions of these survival times. (**K**) The survival time of size-matched SP+ spines was significantly higher than that of SP- spines; computational model; ***p<0.0001, Mann–Whitney U-test. based on 10 samples of 31 SP+ and 31 SP- spines.

The online version of this article includes the following figure supplement(s) for figure 4:

**Figure supplement 1.** Survival curve of spines gaining SP at the beginning of the observation time.

## Synaptopodin-deficiency alters spine survival

We then turned to SP-deficient mice to study spine survival in the absence of SP. (*Figure 6*). First, we compared the distribution of spines of the eGFP-SP-tg and the SP-KO mice and found comparable fractions of spines in the four size categories (*Figure 6A*). This is in line with our observation that SP-deficiency does not affect average spine head size or spine head distribution in vivo (*Figure 1C, D*) or in vitro (*Figure 2E,F*). Next, we compared SP-deficient spines to SP- spines and found that these two groups of spines behave very similar with regard to their survival times and decay kinetics in each of the four size groups (*Figure 6B–R*): survival times were similar and all curves followed single exponential decay functions. This demonstrates that the 'spine phenotype' associated with SP-deficiency does not result in altered spine geometry but in altered spine survival, that is spine stability.

We then wondered how the lack of the stable SP+ spine population affects spine dynamics, in particular spine turnover, of the SP-deficient animals. We reasoned that the reduced stability of large spines will have to be homeostatically compensated by an increased formation of spines and an increased spine turnover. Indeed, these dynamic parameters were significantly increased (*Figure 6—figure supplement 1*), revealing a yet unknown and most likely compensatory phenotype of the SP-deficient mutants.

## Spines shrink and pass through a SP-negative stage before pruning

After investigating the survival of individual spines considering only their initial SP content (SP+ or (SP-); previous paragraphs), we determined changes of the SP+ and SP- spine populations with regard to survival, head size and SP content (*Figure 7*). For this, all spines were identified on day 0 as being either (SP+) or (SP-). For each of the following observation time points, these spines were categorized as either SP+, SP- or 'lost' (*Figure 7A,B*). In addition, the spine head sizes of the surviving SP+ (*Figure 7E*) and SP- (*Figure 7F*) spines were analyzed for each time point. Spines that repeatedly changed their SP content were excluded and analyzed separately (see below). This analysis revealed that in the group of initially SP+ spines the fraction of SP+ spines decreased with time, whereas the fraction of SP- spines increased. This was followed by an increase in the fraction of lost spines (*Figure 7A*). In the group of the initially SP- spines the fraction of surviving spines decreased much more rapidly (*Figure 7B*).

At the observation endpoint (day 12), ~59% of spines which were initially SP+ survived (*Figure 7C*), whereas only ~33% of spines which were initially SP- were still present (*Figure 7D*). The

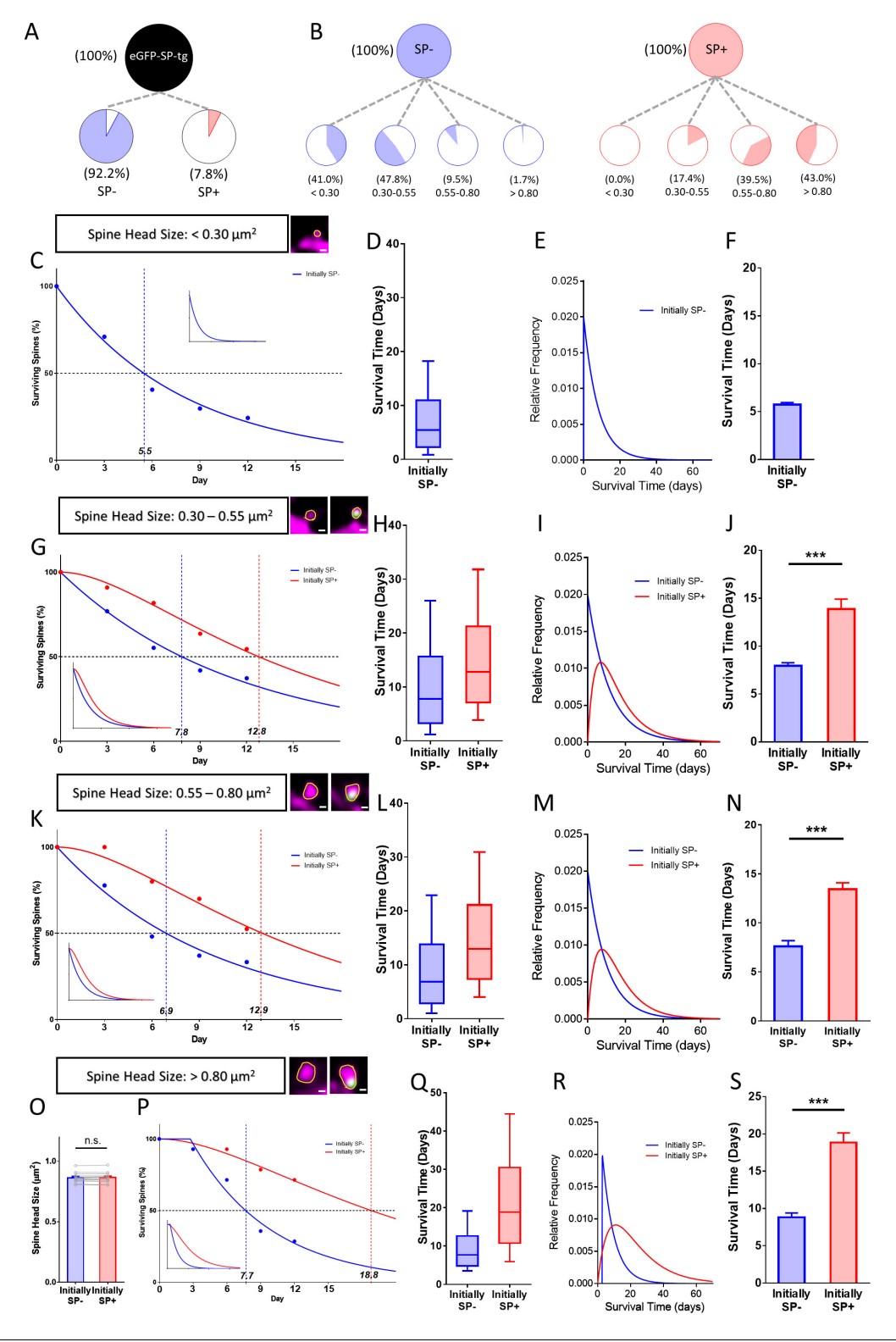

**Figure 5.** Presence of SP in spines correlates with their long-term survival. (**A**) Fraction of SP- (≈ 92.2%) and SP+ spines (≈ 7.8%) in eGFP-SP-tg granule cells on day 0. Number of SP+ spines = 87, SP- spines = 1021, obtained from entire imaged segments; number of cultures: eGFP-SP-tg=24. (**B**) SP- (blue) and SP+ spines (red) were divided into four size classes: very small (<0.3 μm²), small (0.3–0.55 μm²), medium (0.55–0.8 μm²) and large (>0.8 μm²) spines. Whereas most SP- spines were found in the very small (41.0%) and small (47.8%) spine groups, the

*Figure 5 continued on next page*

*Figure 5 continued*

majority of SP+ spines were in the medium (39.5%) and large (43.0%) size groups. (C) Survival curve of very small spines (example shown) from day 0 until day 12. Dots: observed fractions; curves: fitted to observation points. SP- spines (blue line) followed a single exponential decay curve (median survival ≈ 5.5 days). Inset: extrapolated decay curve until day 70. Number of very small SP- spines = 148. (D) Survival time of the total population of SP- spines derived from the fitted survival curve. Box indicates median, lower and upper quartiles. Whiskers indicate 10th and 90th percentiles, respectively. (E) Relative frequency distribution of very small SP- spines. (F) Mean survival time of very small SP- spines; computational model; 10 samples, 148 SP- spines. (G) Survival curves of small SP- and SP+ spines (examples shown). SP- spines (blue line; median survival ≈ 7.8 days) followed a single exponential decay curve, SP+ spines (red line; median survival ≈ 12.8 days) followed a conditional two-stage decay curve. Inset: extrapolated decay curves until day 70. Number of SP- spines = 217; SP+ spines = 11. (H) Survival times and (I) relative frequency distributions of small SP- and SP+ spines. (J) Mean survival time of SP+ spines compared to SP- spines; computational model. ***p=0.0003, Mann–Whitney U-test. 10 samples; 11 SP+ and 217 SP- spines. (K) Survival curves of medium-sized SP- and SP+ spines (examples shown). SP- spines (blue line; median survival ≈ 6.9 days) followed a single exponential decay curve, SP+ spines (red line; median survival ≈ 12.9 days) followed a conditional two-stage decay curve. Inset: extrapolated decay curves until day 70. Number of SP- spines = 27; SP+ spines = 40. (L) Survival times and (M) relative frequency distributions of medium SP- and SP+ spines. (N) Mean survival time of SP+ spines compared to SP- spines; computational model. ***p<0.0001, Mann–Whitney U-test. 10 samples; 40 SP+ and 27 SP- spines. (O) SP+ and SP- spines were size-matched in the largest spine group (mean spine head sizes; not significant; p=0.475, Wilcoxon matched-pairs signed rank test). Number of size-matched pairs n = 14. (P) Survival curves of size-matched large SP- and SP+ spines (examples shown). SP- spines (blue line; median survival ≈ 7.7 days) followed a single exponential decay curve, SP+ spines (red line; median survival ≈ 18.8 days) followed a conditional two-stage decay curve. Inset: calculated decay curves until day 70. (Q) Survival times and (R) relative frequency distributions of large SP- and SP+ spines. (S) Mean survival time of SP+ spines compared to SP- spines; computational model. ***p<0.0001, Mann–Whitney U-test. 10 samples; 14 SP+ and 14 SP- spines.

size of the surviving SP+ (*Figure 7E*) and SP- (*Figure 7F*) spines did not change significantly during the two-week observation period, in line with our finding that spine head size per se is not the major determinant of spine stability. We then looked specifically at the spines that were lost, since our analysis had revealed a shift from SP+ spines to SP- spines (*Figure 7A*). This shift suggested that SP + spines go through a SP- state before pruning. Since we followed the fate and the SP content of every spine over the two-week imaging time period (*Figure 7—figure supplements 1* and *2*), we could test this hypothesis. First, we determined the SP content of the 16% of spines that were lost within one day, that is between days −1 and 0. All of these pruned spines were SP-, in line with our hypothesis (*Figure 7G*). Next, we predicted that SP+ spines should not be lost without going through a SP- state. For this analysis, we looked at all spines that were initially SP+, that is spines SP + at day 0, and focused on those SP+ spines that were pruned sometime during the observation period. Of these spines ~ 94% were SP- prior to their pruning (*Figure 7H*). In the rare cases (n = 2;~6%), in which a SP+ spine seemingly disappeared without passing through a SP- state, the two observation time points were several days apart (e.g., between day 0–3; *Figure 7—figure supplement 1*), making it highly likely that we missed a transitory SP- state of these spines prior to their pruning.

## Loss of SP from spines does not cause pruning

After finding the above evidence that SP is removed from spines prior to their pruning, we wondered whether removal of SP from a spine is inexorably followed by the loss of this spine. If this were the case, all SP+ spines losing their SP cluster should disappear. This hypothesis could be falsified since we observed several SP+ spines (n = 10) undulating between a SP+ and SP- state (*Figure 7I*). Thus, SP can be removed from a spine and reintroduced into a spine without the spine being pruned. Of note, the changes in SP content were accompanied by bidirectional changes in head size (*Figure 7I*, *Figure 2—figure supplement 3*). We also observed SP- spines, which became SP+, lost SP, and became SP+ again at a later stage (n = 8; data not shown). Taken together, we conclude from these observations that loss of a SP cluster from a spine does not cause pruning of that spine.

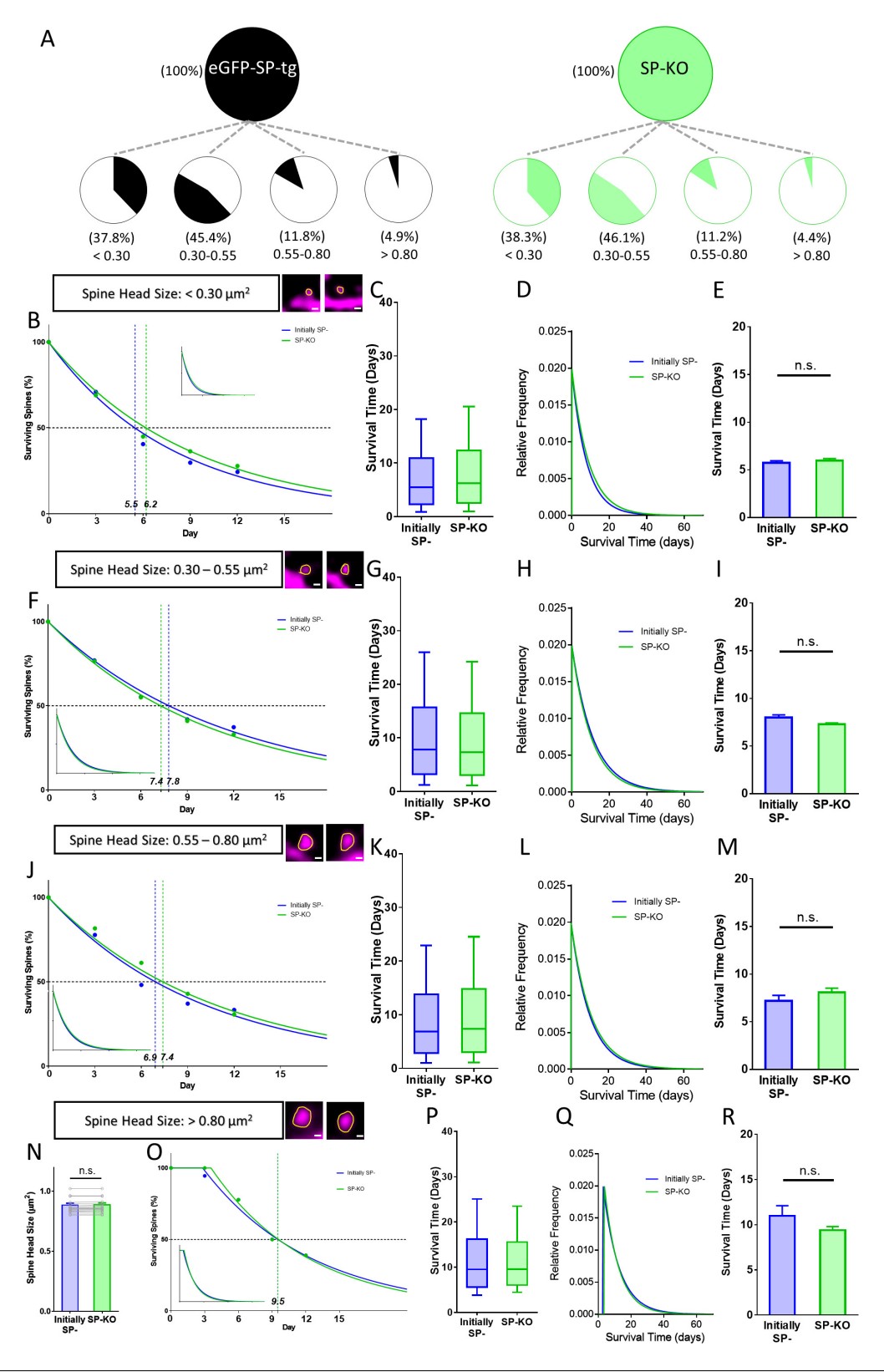

**Figure 6.** Spines of SP-deficient mice decay like SP- spines. (**A**) Spine head size distributions of granule cell dendrites of eGFP-SP-tg and SP-KO mice were similar. Number of spines: eGFP-SP-tg=1107; SP-KO = 1113, obtained from entire imaged segments; number of cultures: eGFP-SP-tg=24; SP-KO = 21. (**B**) Survival curve of very

*Figure 6 continued on next page*

*Figure 6 continued*

small spines (examples shown) from day 0 until day 12. Dots: observed fractions; curves: fitted to observation points. SP- spines (blue line; median survival ≈ 5.5 days) and spines of SP-KO-mice (green line; median survival ≈ 6.2 days) followed single exponential decay curves. Inset: calculated decay curves until day 70. Number of SP-spines = 148; SP-KO spines = 187. (C) Calculated survival times and (D) relative frequency distributions of very small SP- and SP-KO spines. (E) Mean survival time of very small SP- and SP-KO spines; computational model. p=0.529, Mann–Whitney U-test. 10 samples; 148 SP-, 187 SP-KO spines. (F) Survival curves, (G) calculated survival times, and (H) relative frequency distributions of small SP- and SP-KO spines. SP- spines (blue; median survival ≈ 7.8 days) and SP-KO spines (green, median survival ≈ 7.4 days) followed single exponential decay curves. Number of SP- spines = 217; SP-KO spines = 200. (I) Mean survival times; computational model. p=0.089, Mann–Whitney U-test. 10 samples; 217 SP-, 200 SP-KO spines. (J) Survival curves, (K) calculated survival times, and (L) relative frequency distributions of medium-sized SP- and SP-KO spines. SP- spines (blue; median survival ≈ 6.9 days) and SP-KO spines (green, median survival ≈ 7.4 days) followed single exponential decay curves. Number of SP-spines = 27; SP-KO spines = 49. (M) Mean survival times; computational model. p=0.280, Mann–Whitney U-test. 10 samples; 27 SP-, 49 SP-KO spines. (N) SP-KO and SP- spines were size-matched in the largest spine group mean spine head sizes; not significant, p=0.109, Wilcoxon matched-pairs signed rank test. Number of size-matched pairs n = 18. (O) Survival curves, (P) calculated survival times, and (Q) relative frequency distributions of large SP- and SP-KO spines. SP- spines (blue; median survival ≈ 9.5 days) and SP-KO spines (green, median survival ≈ 9.5 days) followed single exponential decay curves. Number of size-matched pairs = 18. (R) Mean survival times; computational model. p=0.393, Mann–Whitney U-test. 10 samples; 18 SP- and 18 SP-KO spines.

The online version of this article includes the following figure supplement(s) for figure 6:

**Figure supplement 1.** SP-KO mice compensate for the loss of SP with an increased spine formation and an increased turnover ratio.

## SP+ spines are more stable than SP- spines even after denervation

Finally, we tested whether the presence of SP within a spine is still correlated with higher spine stability after spines have lost their afferent innervation. Such spines lack glutamatergic input, presynaptic signaling and cell adhesion molecules, which could stabilize a spine or tether it to its innervating terminal (*Yoshihara et al., 2009*). To address this question, we transected the entorhinal axons, that is the so-called perforant path, innervating the distal two-thirds of the dendritic tree of dentate granule cells (*Figure 8A*; *Del Turco and Deller, 2007*). This results in a profound destabilization of denervated granule cell spines (*Vlachos et al., 2012*; *Vlachos et al., 2013a*), making it possible to study the effect of SP remaining within the postsynaptic compartment on spine stability.

Following transection of the entorhinal axons, time-lapse imaging was employed (*Figure 8B*) to follow single identified SP+ spines (*Figure 8C*) and SP- spines (*Figure 8D*) located within the denervation zone (*Vlachos et al., 2012*; *Del Turco et al., 2019*). Survival of the denervated spine population was compared to survival of SP+ and SP- spines from the non-denervated cultures. As shown previously (*Vlachos et al., 2012*), denervation greatly accelerated spine loss (*Figure 8C,D*). This increase in spine loss was observed for both SP+ as well as SP- spines: In the case of the SP+ spines, median survival dropped from 17.5 days in controls to 6.5 days following denervation (*Figure 8C*). Similarly, median survival of SP- spines dropped from 6.8 days in controls to 3.4 days following denervation (*Figure 8D*). As in controls, denervated SP- spines followed a single-phase exponential decay function and SP+ spines followed a conditional two-stage exponential decay function.

Based on these functions we calculated the population distributions (*Figure 8E,F*) and tested – using a data-driven simulation of spine survival – differences in survival times between non-denervated and denervated SP+ and SP- spines, respectively (*Figure 8G,H*). Finally, we compared denervated SP+ with denervated SP- spines (*Figure 8I–M*). To control for any effects of spine head size, size-matched spines were used (*Figure 8I*). Although denervation greatly accelerated the decay of both groups, SP+ spines were still significantly more stable than their SP- size-matched counterparts. From these data we conclude that the presence of SP within the postsynaptic spine compartment is still associated with higher spine stability, even in the absence of a presynaptic element.

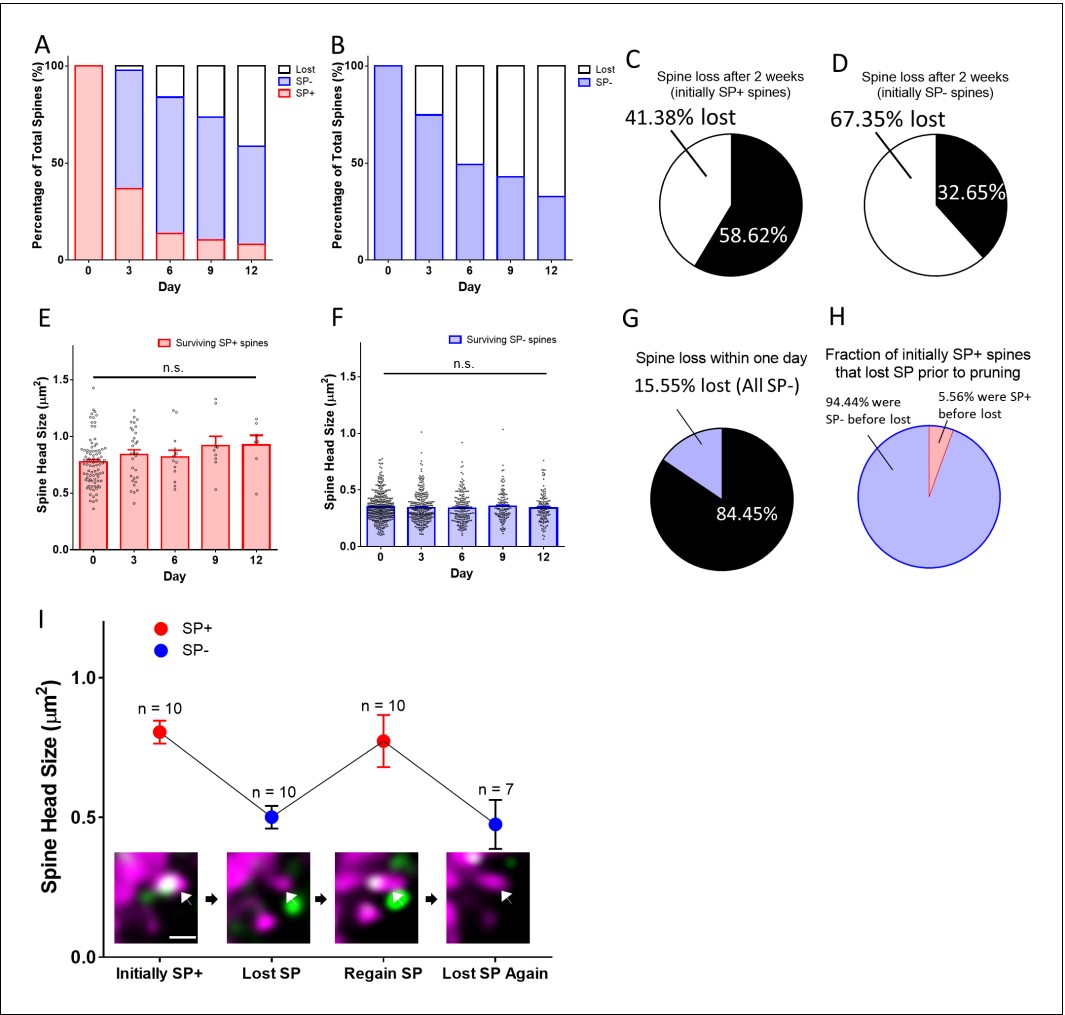

**Figure 7.** Spines pass through a SP-negative state before pruning. (**A, B**) The fates of (**A**) a group of SP+ spines (n = 87) and (**B**) a group of SP- spines (n = 392) are illustrated. SP+ spines gradually become SP- or disappear. SP- spines disappear with time. (**C, D**) Fractions of (**C**) SP+ and (**D**) SP- spines surviving for two weeks. (**E, F**) Mean spine head sizes of surviving (**E**) SP+ and (**F**) SP- spines. Spine head sizes were not significantly different (SP+ spines, p=0.115; SP- spines, p=0.627, Kruskal-Wallis test). (**G**) Analysis of SP content of all spines lost within one day (days −1 to 0; n = 72; 15.55% of 463 spines). All pruned spines were SP-. (**H**) Analysis of all SP+ spines that were pruned during the observation period. Most spines (n = 34) went through a SP- state before disappearing, a minority (n = 2) was lost directly between imaging days 0 and 3. (**I**) A subpopulation of SP+ spines undulated between SP+ and SP- states (n = 10). The gain or loss of SP was accompanied by an increase or decrease of spine head size, respectively. An example for such a case is illustrated (arrow in insets). Scale bar = 1 μm; Single plane deconvolved 2-photon images.

The online version of this article includes the following figure supplement(s) for figure 7:

**Figure supplement 1.** 2-photon time-lapse imaging data of individual SP+ spines.
**Figure supplement 2.** 2-photon time-lapse imaging data of individual SP- spines.
**Figure supplement 3.** 2-photon time-lapse imaging data of individual spines undulating between SP+ and SP-states.

## Mathematical simulations of spine survival are compatible with a conditional two-stage decay process

The decay curves of SP- and SP+ spines (*Figures 4* and *5*) are non-overlapping and mathematically distinct, since they cannot be trivially transformed into each other, for example using a scaling factor. We first analyzed the survival curve of SP- spines, which we could readily simulate and fit to our data using a single-phase exponential decay function:

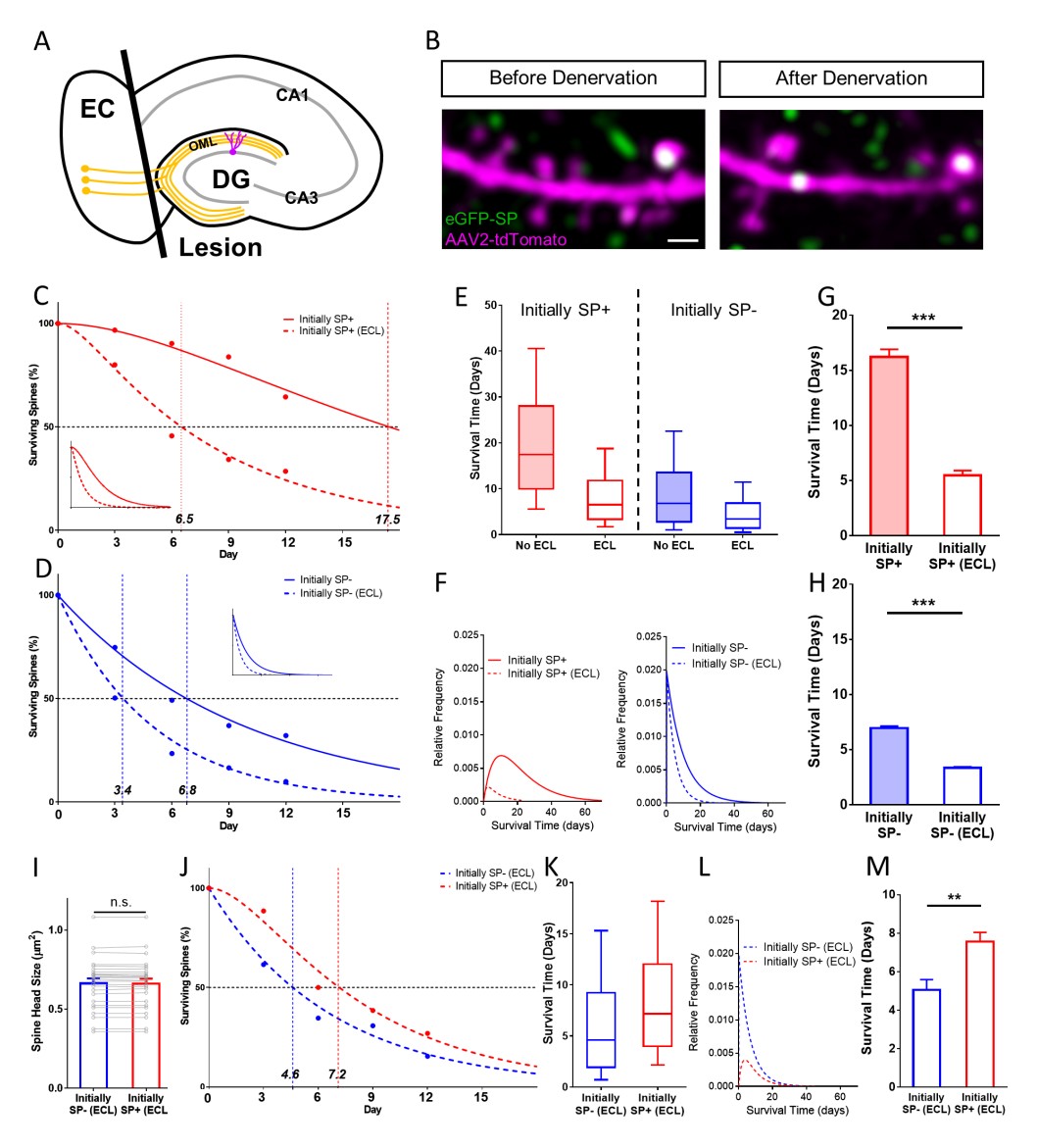

**Figure 8.** SP+ spines are more stable than SP- spines after denervation. (**A**) Schematic of the entorhinal denervation in vitro model. The entorhinal projection (perforant path; orange) is transected under visual control and granule cell dendrites extending into the outer molecular layer (OML) of the dentate gyrus (DG) are denervated. CA3, CA1, hippocampal subfields; EC, entorhinal cortex. (**B**) 2-photon images of a granule cell dendrite in the OML before and 4 days after denervation. Note the denervation-induced spine loss. Scale bar = 1 μm. (**C, D**) Survival curve of SP+ (**C**) and SP- (**D**) spines in denervated cultures compared to SP+ and SP- spines in cultures without lesion (control groups also shown in *Figure 4*). The median survival time of both spine populations is reduced after denervation. SP+ spines still followed a sequential two-stage decay function after denervation, SP- spines still followed a single-phase exponential decay kinetic. Inset: Projected decay curves fitted to the observed survival fractions. Number of SP+ spines at day 0: denervated = 35, non-denervated = 31; Number of SP- spines at day 0: denervated = 302, non-denervated = 392. (**E**) Both SP+ (≈ 6.5 days) and SP- (≈ 3.4 days) spines had a reduced median survival time following denervation. (**F**) Relative frequency distributions of median survival times of SP+ and SP- spines derived from the fitted survival curves. (**G, H**) The mean survival time of SP+ (**G**) and SP- (**H**) spines was significantly reduced after denervation; computational model. ***p<0.0001, Mann–Whitney U-test. 10 samples of size; 35 SP+ spines with denervation and 31 SP+ spines without denervation; 302 SP- spines with denervation and 392 SP- spines without denervation. (**I**) To control for differences in spine head size, SP+ and SP- spines of equal head size were matched. The mean spine head size of these pairs was not significantly different. p=0.670, Wilcoxon matched-pairs signed rank test. Number of size-matched pairs = 26. (**J**) The decay curves of size-matched spines were similar to the decay curves of all SP+ and SP- spines after

*Figure 8 continued on next page*

*Figure 8 continued*

denervation. The half-life of SP- spines was ≈ 4.6 days while SP+ spines had a half-life of ≈ 7.2 days. (**K, L**) Calculated median survival times (**K**) and relative frequency distributions (**L**) of size-matched SP+ and SP- spines. (**M**) The mean survival time of size-matched SP+ spines was significantly higher than SP- spines. Computational model. **p=0.0021, Mann–Whitney U-test. 10 samples of size 26 SP+ and 26 SP- spines.

$$Y = 100\% * \exp\left(-\frac{t - offset}{\tau spine}\right)$$

In this equation, the variable Y denotes the percentage of SP- spines surviving up to day t starting with 100% SP- spines present at day zero. τspine denotes the decay time constant of spine loss. The half-life of spines was determined as 0.69*τspine. 'offset' denotes the start of the decay. Fitting the data to this model, τspine of SP- spines is ~9.8 days and the half-life of these spines is ~6.8 days (*Figure 4*). Size-matched SP- spines (*Figure 4G,H*), SP- spines of different sizes (*Figure 5*), and SP-deficient spines (*Figure 6*) followed similar single-phase exponential decay functions.

In contrast, the survival curve of SP+ spines did not follow a single-phase exponential decay function. It was also distinct from a linear function. An almost perfect fit could be obtained, however, by using a conditional two-stage exponential decay function, identical to the one used to model the conditional two-stage decay of radioisotopes (*Bateman, 1910*). In this model, a radioisotope undergoes decay and forms an unstable intermediary. This intermediary decays again into a stable isotope. The two decay processes have different decay time constants. We now applied this function to the data obtained for spine survival: SP+ spines were considered to lose SP with the time constant τSP. The new SP- spine would then disappear with the time constant τspine, that is the same time constant determined for the SP- spines above:

$$Y = 100\% * \left(\frac{\exp\left(\frac{-t}{\tau spine}\right)}{\tau SP * \left(\frac{1}{\tau SP} - \frac{1}{\tau spine}\right)} + \frac{\exp\left(\frac{-t}{\tau SP}\right)}{\tau SP * \left(\frac{1}{\tau spine} - \frac{1}{\tau SP}\right)} + \exp\left(\frac{-t}{\tau SP}\right)\right)$$

In this equation, the variable Y denotes the percentage of SP+ spines surviving up to day t starting with 100% of SP+ spines present at day zero. We fitted the survival data for SP+ spines to this equation and found the time constant τSP of the first decay process (SP+ spines becoming SP-) to be ~11.1 days and accordingly the half-life of SP in spines to be ~7.7 days. Thus, after approximately 17.4 days, ~50% of all initially SP+ spines would have disappeared (*Figure 4*). Size-matched SP+ spines (*Figure 4G,H*), SP+ spines of different sizes (*Figure 5*) and SP+ spines after denervation (*Figure 8C,J*) followed similar conditional two-stage exponential decay functions.

We conclude from this mathematical analysis of our experimental data that the survival curves of SP+ spines are compatible with a conditional two-stage decay process in which SP+ spines first lose their SP cluster (τSP) before being subsequently pruned as SP- spines with the time constant (τspine; same as initially SP- spines; *Figure 9*).

## Discussion

Spine geometry and spine stability are structural parameters that have been linked to synaptic strength, network reorganization, behavioral learning, and memory trace formation (*Bourne and Harris, 2007*; *Kasai et al., 2010*; *McKinney, 2010*; *Roberts et al., 2010*; *Koleske, 2013*; *Rogerson et al., 2014*; *Segal, 2017*). A molecule potentially involved in these biological phenomena is the actin-modulating protein SP (*Mundel et al., 1997*; *Deller et al., 2000a*; *Jedlicka and Deller, 2017*). Using a combination of mouse genetics, viral transduction and 2-photon time-lapse imaging, the effects of SP on spine geometry and stability were analyzed for two weeks in an organotypic environment. Our major findings can be summarized as follows: (1) SP is primarily present in large spines. (2) SP+ spines of all sizes are more stable than SP- spines. (3) Spines of SP-deficient animals decay like SP- spines. In these mutants, a compensatory increase in spine turnover is observed. (4) SP+ spines that later underwent pruning passed through a SP- state. These spines followed a conditional two-stage decay kinetic. (5) SP+ spines are more stable than SP- spines following denervation.

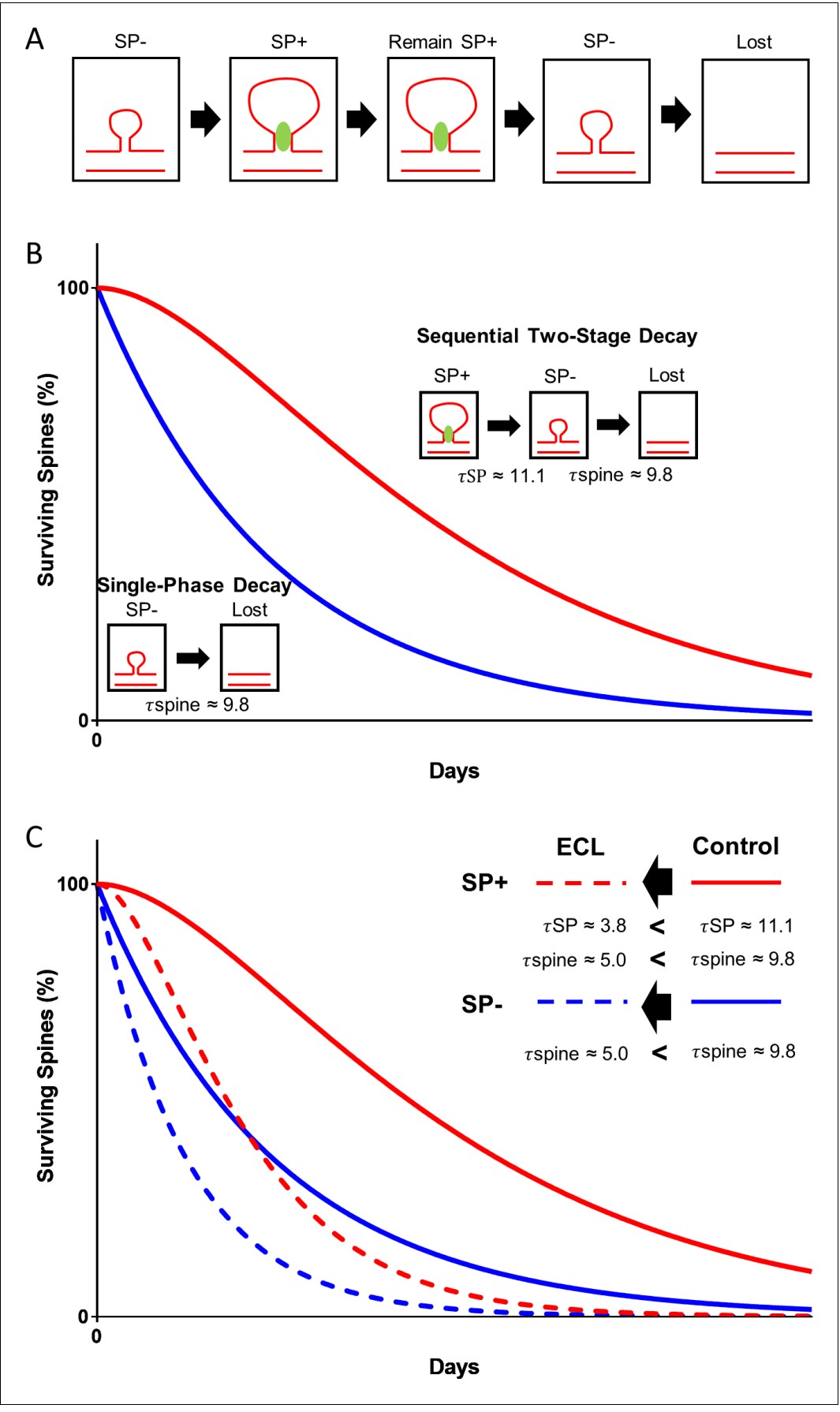

**Figure 9.** Model illustrating the role of SP in spine head size and spine stability. (**A**) SP- spines gaining SP show a concomitant increase in spine head size. SP in spines stabilizes spines. Prior to spine loss, SP is removed from spines. Loss of SP from spines is associated with a reduction in spine head size. (**B**) The data of the present study suggest a conditional two-stage exponential decay model for SP+ spines. The first stage involves SP+ spines losing SP (τSP) and the second stage involves the gradual disappearance of these SP- spines (τspine). For SP- spines, our data suggest they undergo a classical single-phase exponential decay model where spines are lost exponentially with time. (**C**) Following denervation, the two survival curves are shifted to the left. However, even under these conditions, SP+ spines are more stable than SP- spines and follow a conditional two-stage exponential decay kinetic. Entorhinal cortex lesion, ECL.

Together, our results implicate SP as a major regulator of spine stability. Since SP is present in ~65% of large (>0.8 µm$^2$) spines, its presence in these spines could explain their higher stability (e.g., *Matsuzaki et al., 2004*; *Kasai et al., 2010*; *McKinney, 2010*).

## The presence of SP rather than spine head size is associated with spine stability

The biological functions of SP in spines have not been fully understood. After effects of SP on spine density and spine length were ruled out (*Deller et al., 2003*), roles in spine head expansion (*Zhang et al., 2013*) and in the maintenance of an expanded spine head (*Okubo-Suzuki et al., 2008*; *Konietzny et al., 2019*) were suggested. In our study, we could confirm the association between SP and spine head size in an organotypic environment and asked how SP affects spines over longer time periods. Since SP is tightly associated with large spines and since these spines are believed to be more stable than small spines (*Matsuzaki et al., 2004*; *Kasai et al., 2010*), we wondered whether SP could be the 'molecular longevity factor' within this subgroup of stable spines. Our imaging data suggest that SP could indeed play such a role and that it is not the size of a spine per se that is important for stability but rather the presence of SP within a spine. This conclusion is supported by the following observations: (1) Small spines (0.3–0.55 µm$^2$ head area) containing SP have a long lifetime and a median survival of 12.8 days. Large spines (>0.8 µm$^2$ head area) without SP have a median survival time of 7.7 days. Thus, small SP+ spines are more stable than large SP- spines. (2) Comparison of the median survival times of SP- spines revealed that their median survival time was similar among all size groups (i.e., very small, small, medium and large) and much shorter than the median survival time of SP+ spines of any size. Finally, (3) all spines, including the large spines, of SP-deficient animals decayed like SP- spines and had short median survival times. Although this does not fully exclude a small effect of spine size on spine survival, for example large spines may need time to shrink prior to their removal (see next paragraph), our data show that the presence of SP within a spine is a much more reliable indicator of its stability than the size of its head.

## SP+ spines that are pruned pass through a SP- state and shrink in size

Two weeks of time-lapse imaging of identified SP+ spines made it possible to study their fate: Single SP+ spines surviving for the entire time period, SP+ spines losing and regaining SP, as well as SP+ spines that were pruned could be identified and analyzed (*Figure 7—figure supplements 1–3*). From these multidimensional time-lapse data, we could draw additional conclusions: First, from the fact that some spines lose and regain SP with time, we concluded that loss of SP from a spine is not by itself sufficient to tag a spine for pruning. Rather, such spines seem to re-enter the large pool of SP- spines from which new SP+ spines can be recruited. Second, by following SP+ spines that were pruned during the observation period, we could show that in almost all cases, these spines went through a SP- state before they disappeared. Thus, our data suggest that removal of the spine stabilizing protein SP is a necessary step before the actin cytoskeleton of a spine can be degraded and the spine can be pruned. Third, in line with the fact that SP is associated with large spines, we observed that spines losing SP also shrank in size (*Figure 7—figure supplements 1–3*). In the case of spines changing their SP content (*Figure 7—figure supplement 3*) this also resulted in corresponding head size changes. In agreement with these observations, we found that SP+ spines designated to be pruned follow a conditional two-stage decay function, similar to the one described for radioactive isotopes in Physics (*Bateman, 1910*): The first stage is the slow 'decay' of SP+ spines

into a SP- state. The second stage is the pruning of these SP- spines, which goes as fast as for initially SP- spines (*Figure 9*).

## SP-deficient neurons homeostatically compensate for the loss of SP with increased spine formation

The role of SP in the regulation of spine stability is further supported by data from SP-deficient mice. Granule cells of these mice exhibited spines that decayed like SP- spines. In addition, SP-deficient granule cells compensated for the reduced stability of spines by increasing their spine formation and thereby their spine turnover ratio. This finding suggests that SP-deficient neurons homeostatically compensate for the loss of stability by upregulating their spine formation. This may explain why loss or overexpression of SP does not seem to affect spine densities (*Deller et al., 2003*; *Okubo-Suzuki et al., 2008*; *Vlachos et al., 2009*). Whereas spine density appears to be regulated by other molecules, including among others hormones (*Woolley, 2000*; *Segal and Murphy, 2001*; *Fester et al., 2012*; *Luine and Frankfurt, 2012*), actin-modulating proteins (*Yamazaki et al., 2014*), Fragile X mental retardation protein (*Irwin et al., 2000*; *Bagni and Zukin, 2019*), non-coding RNAs (*Briz et al., 2017*), neurotrophic factors (*Murphy et al., 1998*) and adhesion molecules (*Segura et al., 2007*; *Keeler et al., 2015*; *Müller et al., 2017*), SP appears to be a major regulator of spine stability.

## Interactions of SP with the spine cytoskeleton may increase the stability of spines

How could SP exert its stabilizing effects on spines? In comparison to other well-established stabilizers of spines, such as PSD-95 (*El-Husseini et al., 2000*; *Cane et al., 2014*), which primarily act at the synapse (*Ehrlich et al., 2007*; *De Roo et al., 2008*; *Yoshihara et al., 2009*; *Meyer et al., 2014*), the preferential localization of SP clusters in the lower part of the spine head argues for an association of SP with the stable pool of actin, which forms the central and less dynamic core of spines (*Bramham et al., 2008*; *Bosch et al., 2014*; *Colgan and Yasuda, 2014*; *Spence and Soderling, 2015*). Extant data suggest that a stabilizing effect of SP on actin can be caused by several direct and indirect interactions: (i) SP directly binds to actin and can induce actin aggregates (*Mundel et al., 1997*; *Asanuma et al., 2005*), (ii) SP interacts with alpha-actinin-2 and in the presence of both proteins long, unbranched and highly stable actin filaments are formed (*Asanuma et al., 2005*), (iii) SP protects actin filaments from disruption (*Okubo-Suzuki et al., 2008*; *Wang et al., 2016*), potentially by blocking signals inducing actin fragmentation. For kidney podocytes, interactions of SP with Rho-GTPase family members have been reported (*Asanuma et al., 2006*; *Yanagida-Asanuma et al., 2007*). In neurons, Rho-GTPases regulate the maintenance of spines and RhoA in particular promotes prolonged spine enlargement (*Tashiro and Yuste, 2008*; *Murakoshi et al., 2011*). By protecting RhoA from proteasomal degradation (*Asanuma et al., 2006*), SP could increase spine stability. (iv) Finally, SP may stabilize spines indirectly via the spine apparatus organelle and its function as a $Ca^{2+}$ store (*Korkotian and Segal, 2011*). Higher $Ca^{2+}$ transients could stabilize spines via the postsynaptic $Ca^{2+}$-sensor caldendrin (*Mikhaylova et al., 2018*), or via activation of kinases, which, in turn, inactivate actin-depolymerizing proteins such as cofilin (see *Borovac et al., 2018*, for review). Of note, cofilin (*Bosch et al., 2014*; *Borovac et al., 2018*) and SP (this study; *Bas Orth et al., 2005*) are found in similar locations within the spine compartment, hinting at some kind of crosstalk between these proteins. In sum, SP could influence the stability of actin pools and/or actin treadmilling in spines via several mutually non-exclusive direct and indirect mechanism.

## Implications for synaptic plasticity and long-term memory

SP plays an important role in Hebbian- (*Yamazaki et al., 2001*; *Deller et al., 2003*; *Okubo-Suzuki et al., 2008*; *Holbro et al., 2009*; *Jedlicka et al., 2009*; *Vlachos et al., 2009*; *Zhang et al., 2013*; *Jedlicka and Deller, 2017*) and homeostatic (*Vlachos et al., 2013b*) forms of synaptic plasticity. Under conditions of synaptic strengthening SP is upregulated (*Yamazaki et al., 2001*), sorted within neurons (*Fukazawa et al., 2003*) and recruited into activated spines (*Okubo-Suzuki et al., 2008*; *Vlachos et al., 2009*). In the context of Hebbian plasticity, SP appears to be part of the downstream machinery executing functional and structural changes (see *Jedlicka and Deller, 2017*, for

review). At the functional level, SP plays an important role as an essential molecule of the spine apparatus organelle, which acts as a local calcium source and sink (*Vlachos et al., 2009*; *Korkotian and Segal, 2011*; *Korkotian et al., 2014*) and which regulates AMPA-R trafficking to excitatory postsynapses (*Vlachos et al., 2009*). At the structural level, it is recruited to enlarged spines (*Okubo-Suzuki et al., 2008*; *Vlachos et al., 2009*; *Zhang et al., 2013*) and increases their stability (this study). Since SP+ spines are large and strong spines exhibiting a high density of AMPA-R (*Vlachos et al., 2009*), loss of SP from only this fraction of spines suffices to impair Hebbian plasticity (*Deller et al., 2003*; *Jedlicka et al., 2009*; *Vlachos et al., 2009*; *Zhang et al., 2013*; *Grigoryan and Segal, 2016*), underlining its role in Hebbian forms of synaptic strengthening.

At the behavioral level, learning has been linked to spine stability: Prior to learning, spines are dynamic and rapidly turned over. After instructive experience, however, spines are more stable and their synapses have been strengthened (*Roberts et al., 2010*). Similarly, at the network level, neuronal ensemble formation and long-term memory require strong and stable synapses (*Kasai et al., 2010*; *Koleske, 2013*; *Rogerson et al., 2014*; *Segal, 2017*; *Yokose et al., 2017*). In the present study, we have linked the stability of spines to the presence of the actin-modulating protein SP. This makes it attractive to speculate that synapses requiring a high degree of stability, for example synapses after a learning experience (*Roberts et al., 2010*) or synapses binding excitatory neurons together into functional ensembles (*Yokose et al., 2017*), require SP as their stabilizer. Conversely, a reduction in spine stability has been reported in aged (*Voglewede et al., 2019*) and pathologically altered brains (*Halpain et al., 2005*; *Spires-Jones et al., 2007*). Since SP is reduced under these conditions (*Reddy et al., 2005*; *Counts et al., 2014*; *Sidhu et al., 2016*; *Wingo et al., 2019*), it is conceivable that insufficient levels of SP could contribute to brain pathologies in which network stability is reduced.

# Materials and methods

**Key resources table**

| Reagent type (species) or resource | Designation | Source or reference | Identifiers | Additional information |
|---|---|---|---|---|
| Strain, strain background (*Mus musculus*) | eGFP-SP-tg | *Vlachos et al., 2013b* | | C57BL/6J background |
| Strain, strain background (*Mus musculus*) | SP-KO | *Deller et al., 2003* | RRID:IMSR_JAX:028822 | C57BL/6J background |
| Strain, strain background (*Mus musculus*) | WT | Janvier Labs | | C57BL/6J background |
| Transfected construct (*Homo sapiens*) | phSyn-tdTomato | *Radic et al., 2017* | | |
| Antibody | Rabbit Polyclonal Anti-Synaptopodin | Synaptic Systems | RRID:AB_887825 | (1:1000) |
| Antibody | Goat Anti-Rabbit IgG Antibody (H+L), Biotinylated | Vector Laboratories | RRID:AB_2313606 | (1:200) |
| Recombinant DNA reagent | in situ mRNA probe: Synaptopodin | *Mundel et al., 1997* | | (1:500) |
| Commercial assay or kit | RNeasy Plus Micro Kit | Qiagen | Cat #74034 | |
| Commercial assay or kit | High Capacity cDNA Reverse Transcription Reagents Kit | Applied Biosystems | Cat #4368814 | |
| Commercial assay or kit | TaqMan PreAmp Master Mix | Applied Biosystems | Cat #4391128 | |

*Continued on next page*

*Continued*

| Reagent type (species) or resource | Designation | Source or reference | Identifiers | Additional information |
|---|---|---|---|---|
| Commercial assay or kit | TaqMan 178 Gene Expression Assays | Applied Biosystems | Synpo_B-ANY: Custom Gene Expression Assay, FAM, ID: AIWR2E7 Mouse (Assay by Design, designed by Domenico Del Turco) GAPD (GAPDH) Endogenous Control (FAM/ MGB probe, primer limited) Cat #4352932E | |
| Commercial assay or kit | TaqMan Gene Expression Master Mix | Applied Biosystems | Cat #4369016 | |
| Chemical compound, drug | Alexa Fluor 568 Hydrazide | Invitrogen | Cat #A10437 | 0.75 mM |
| Software, algorithm | FV10-ASW | Olympus | | Image acquisition |
| Software, algorithm | Eclipse C1 plus | Nikon | | Image acquisition |
| Software, algorithm | Scanimage 5.1 | Vidrio Technologies | | Image acquisition |
| Software, algorithm | Huygens Professional Version 17.10 | Scientific Volume Imaging | | Image processing |
| Software, algorithm | Fiji/ImageJ 1.52 hr | National Institutes of Health | | Image processing and data analysis |
| Software, algorithm | ImageSP Viewer | SysProg | | Image processing and data analysis |
| Software, algorithm | GraphPad Prism 6 | GraphPad Software | | Statistical analysis |
| Software, algorithm | LabView 2019 | National Instruments | | Data analysis |

## Animals

Adult male mice (12–34 weeks) lacking SP (SP-KO, C57BL/6J background; *Deller et al., 2003*), wild-type mice (WT, C57BL/6J background) and Thy1-eGFP-SP x SP-KO mice eGFP-SP-tg, C57BL/6J background (*Vlachos et al., 2013b*) were used for the ex vivo analysis of granule cell spines in fixed slices. eGFP-SP-tg and SP-KO mice were used for the preparation of organotypic entorhino-hippocampal tissue cultures. eGFP-SP-tg mice were obtained by cross-breeding homozygote Thy1-eGFP-SP-transgenic mice (*Vlachos et al., 2013a*) with SP-deficient mice (*Deller et al., 2000a*). Thus, eGFP-SP-tg mice used for tissue culture preparation were monoallelic for eGFP-SP and devoid of endogenous SP. Adult mice used for fixed brain tissues were bred and housed at mfd Diagnostics GmbH, Wendelsheim, while mice used for organotypic entorhino-hippocampal tissue cultures were bred and housed at the animal facility of Goethe University Hospital Frankfurt. Animals were maintained on a 12 hr light/dark cycle with food and water available ad libitum. All animal experiments were performed in accordance with the German animal welfare law and had been declared to the Animal Welfare Officer of the Medical Faculty (Wa-2014–35). Every effort was made to minimize the distress and pain of animals.

## Intracellular injections of granule cells in fixed tissue

After delivery, animals were kept in an in-house scantainer for a minimum of 24 hr. Animals were killed with an overdose of intraperitoneal Pentobarbital and subsequently intracardially perfused (0.1 M Phosphate Buffer Saline (PBS) containing 4% paraformaldehyde (PFA)). Tail biopsies were obtained after death to re-confirm the genotype. Brains were taken out immediately after perfusion, post-fixed (18 hr, 4% PFA in 0.1 M PBS, 4° C), washed trice in ice-cold 0.1 M PBS, sectioned (250 µm) on a vibratome (Leica VT 1000 s) and stored at 4°C until use.

Intracellular injections of granule cells in fixed slices were performed as previously described (*Germroth et al., 1989*; *Hick et al., 2015*), with modifications. Hippocampal slices were placed in a custom-built, transparent, and grounded recording chamber filled with ice-cold 0.1 M PBS. The chamber was attached to an epifluorescence microscope (Olympus BX51WI; 10x objective LMPlanFLN10x, NA 0.25, WD 21 mm) mounted on an x-y translation table (Science Products, VT-1 xy Microscope Translator). Sharp quartz-glass microelectrodes (Sutter Instruments, QF100-70-10, with filament) were pulled using a P-2000 laser puller (Sutter Instruments). Microelectrodes were tip-loaded with 0.75 mM Alexa568-Hydrazide (Invitrogen) in HPLC-grade water (VWR Chemicals, HiPer-Solv CHROMANORM) and subsequently back-filled with 0.1 M LiCl in HPLC-grade water. Micro-electrodes were attached to an electrophoretic setup via a silver wire and 500 MΩ resistance. The tip of the microelectrode was navigated into the granule cell layer using a micromanipulator (Märzhäuser Wetzlar, Manipulator DC-3K). A square-wave voltage (1 mV, 1 Hz) was applied using a voltage generator (Gwinstek SFG-2102). Granule cells were filled under visual control for at least 10 min or until no further labeling was observed. Injected sections were fixed (4% PFA in PBS, overnight, 4° C, in darkness), washed in 0.1 M PBS and mounted on slides (Dako fluorescence mounting medium, Dako North America Inc). Only granule cells with dendrites reaching the hippocampal fissure were used for analysis.

## Organotypic tissue cultures

Organotypic entorhino-hippocampal tissue cultures (300 µm thick) were prepared from postnatal 4–5 days old eGFP-SP-tg mice of either sex as previously described with minor modifications (*Del Turco and Deller, 2007*; *Vlachos et al., 2012*). Culture incubation medium contained 42% MEM, 25% Basal Medium Eagle, 25% heat-inactivated normal horse serum, 25 mM HEPES, 0.15% sodium bicarbonate, 0.65% glucose, 0.1 mg/ml streptomycin, 100 U/ml penicillin, 2 mM glutamax, adjusted to pH 7.30. The cultivation medium was refreshed every 2 to 3 days. All tissue cultures were allowed to mature in vitro for at least 25 days in a humidified incubator (5% $CO_2$, at 35°C).

## Transection of entorhinal afferents to the dentate gyrus in tissue cultures

Entorhinal denervation of organotypic tissue cultures was performed as previously described (*Del Turco and Deller, 2007*; *Vlachos et al., 2012*): A subset of tissue cultures was completely transected from the rhinal fissure to the hippocampal fissure using a sterile scalpel blade after the initial imaging session on day −1. To ensure complete and permanent separation of the entorhinal cortex from the hippocampus, the entorhinal cortex was removed from the culturing dish in every denervation experiment.

## Laser microdissection

Tissue cultures used for laser microdissection were washed with 0.1 M PBS, embedded in tissue-freezing medium (Leica Microsystems) and shock-frozen (−80°C) as previously described with minor modifications (*Vlachos et al., 2013a*). Sections (8–10 µm) were cut on a cryostat (Leica CM 3050 s) and mounted on polyethylene terephthalate (PET) foil metal frames (Leica Microsystems). Sections were fixed in acetone (−20°C, 30–45 s), stained with 0.1% toluidine blue (Merck) at room temperature for 20 s before rinsing in ultrapure water (DNase/RNase free, Invitrogen) and dehydrated in ethanol. PET foil metal frames were mounted on a Leica LMD 6000 B system (Leica Microsystems) and the granule cell layer was dissected using a pulsed UV laser beam. In a second experiment using virus-injected eGFP-SP-tg and wildtype cultures, only areas of the granule cell layer injected with the virus were harvested. The micro-dissected material was collected in microcentrifuge tube caps placed underneath the frame. Caps contained guanidine isothiocyanate-containing buffer (RNeasy

Mini Kit, Qiagen) and 1% β-mercaptoethanol (AppliChem GmbH). Micro-dissected tissue samples were stored at −80℃ until further processing.

## RNA extraction, reverse transcription and quantitative polymerase chain reaction

Total RNA was isolated using the RNeasy Plus Micro Kit (Qiagen) and subsequently reverse transcribed using High Capacity cDNA Reverse Transcription Reagents Kit (Applied Biosystems) following the manufacturer's recommendations. Before real-time quantitative polymerase chain reaction (RT-qPCR), cDNA was preamplified using TaqMan PreAmp Master Mix (Applied Biosystems) with a standard amplification protocol of 14 cycles. Quantitative PCR was performed using TaqMan Gene Expression Assays (Applied Biosystems): a custom TaqMan Gene Expression Assay for Synaptopodin (accession number NM_001109975.1; forward primer 5′-GTCTCCTCGAGCCAAGCA-3′; reverse primer 5′-CACACCTGGGCCTCGAT-3′ and probe 5′-TCTCCACCCGGAATGC-3′) and Gapdh (Mm99999915_g1) as a reference gene for normalization. qPCR conditions were carried out using TaqMan Gene Expression Master Mix (Applied Biosystems) with the StepOnePlus Real-Time PCR System (Applied Biosystems). No signals were detected in no-template controls. qPCR-data were analyzed as described by *Pfaffl, 2001*. The qPCR assay efficiency was calculated with the StepOnePlus software (Applied Biosystems, USA) based on a dilution series of 5 samples for each assay.

## Adeno-associated virus (AAV) production

HEK293T cells were transfected with pDP1rs (Plasmid Factory), pDG (Plasmid Factory), and tdTomato-vector plasmid (12:8:5) by calcium phosphate seeding and precipitation (*Grimm et al., 1998*). Cells were collected 48 hr after transfection, washed twice with 0.1 M PBS, centrifuged at 1.500 x g for 5 min and resuspended in 0.1 M PBS. Viral particles within the cells were released by four freeze-thaw cycles and the supernatant centrifuged at 3200 x g for 10 min to remove cell debris. The final supernatant was collected, aliquoted and stored at −80℃.

## Viral labeling

To label dentate granule cells, tissue cultures were transduced with an AAV serotype 2 (AAV2) containing the gene for tdTomato under the human Synapsin one promoter (*Radic et al., 2017*). Local injections were performed on cultures using an injection pipette pulled from thin-walled borosilicate capillaries (Harvard Apparatus, 30–0066). The pipettes were held by a head stage with a HL-U holder (Axon Instruments) and positioned using a micro-manipulator (Luigs and Neumann). Approximately 0.05–0.1 µl of AAV2-hSyn-tdTomato was injected directly into the suprapyramidal blade of the dentate gyrus using a syringe. Tissue cultures were visualized with an upright microscope (Nikon FN1) equipped with a camera and software (TrueChrome Metrics) using a 10x water immersion objective lens (Nikon Plan Fluor, NA 0.30), which allowed precise injection of the AAV2 into the DG. Injections were performed 2–3 days after the tissue cultures were prepared.

## Fluorescence in-situ hybridization

Organotypic tissue cultures (OTC) were fixed at 26 days in vitro (DIV) in 4% paraformaldehyde (PFA) for 30 min at room temperature (RT) followed by an incubation in 30% sucrose overnight at 4℃. Slices of 30 µm were cut with a cryostat (CM3050S, Leica), collected in 2x SSC (LONZA AccuGEN) and subsequently incubated in 10 mM citrate buffer (pH 6) for 20 min at 85℃, washed several times in 2x SSC, and prehybridized for 2 hr at 60℃ with hybridization buffer (50% formamide, 5x SSC, 5% dextran sulfate, 500 µg/ml DNA MB grade from sperm [Roche], 250 µg/ml t-RNA [Sigma–Aldrich] and 1x Denhardt's [Sigma–Aldrich]). After heat treatment of Synaptopodin fluorescence in-situ hybridization (FISH) probe for 5 min at 85℃, sections were incubated with Synaptopodin probe in hybridization buffer (1:500) overnight at 60℃. Several washing steps (2x SSC for 10 min at RT, 2x SSC/50% formamide [AppliChem] for 15 min at 60℃, 0.1x SSC/50% formamide for 15 min at 60℃, 0.1x SSC for 15 min at 60℃, and TN buffer [0.1 M Tris-HCl, 0.15 M NaCl; pH 7.4] for 5 min at RT) were performed before blocking solution (1% blocking reagent [Roche] in TN buffer) was added for 30 min at RT. After blocking, anti-Digoxigenin-POD (1:2000 in blocking solution; Roche) was added to sections for 2 hr at RT. Sections were washed several times with TNT (0.1 M Tris-HCl, 0.15 M NaCl, 0.3% Triton X-100) and incubated in TSA-Plus Cyanine 3 System (Perkin Elmer) diluted in amplification

solution (1:50) for 10 min in the dark at RT. After several washing steps with TNT, sections were counterstained with Hoechst 33342 to visualize nuclei and finally mounted in Fluorescence Mounting Medium (Dako, Agilent Technologies).

## Electron microscopy

Tissue cultures were fixed for 2 hr in 0.1 M sodium cacodylate buffer (CB) containing 4% PFA, 4% sucrose, 15% picric acid and 0.5% glutaraldehyde. After four washes in CB, the fixed cultures were resliced to 40 µm. Following a wash with 0.1 M tris-buffered saline (TBS), free floating sections were treated with 0.1% $NaBH_4$ (Sigma-Aldrich) in TBS for 10 min to remove unbound aldehydes. Sections were then blocked with 5% Bovine Serum Albumin (BSA) in TBS for 1 hr at room temperature.

For detection of SP, sections were first incubated with rabbit anti-SP primary antibody (1:1000; Synaptic Systems) in 2% BSA and 0.1 M TBS for 18 hr at room temperature followed by incubation with biotinylated goat anti-rabbit Immunoglobin G secondary antibody (1:200; Vector Laboratories, Burlingame, CA) in 2% BSA and 0.1 M TBS for 75 min at room temperature. After three washes with TBS, sections were incubated in avidin-biotin-peroxidase complex (ABC-Elite, Vector Laboratories) for 90 min at room temperature and treated with diaminobenzidine solution (Vector Laboratories) for 2–15 min at room temperature. Sections were then incubated for silver-intensification in 3% hexa-methylenetetramine (Sigma-Aldrich), 5% silver nitrate (AppliChem) and 2.5% di-sodium tetraborate (Sigma-Aldrich) for 10 min at 60°C, followed by 0.05% tetrachlorogold solution (AppliChem) for 3 min and 2.5% sodium thiosulfate (Sigma-Aldrich) for 3 min with three washes after each step using distilled water.

After immunostaining, sections were washed in 0.1 M CB, treated with 0.5% $OsO_4$ (Plano) in 0.1 M CB for 30 min, dehydrated in increasing concentrations of ethanol and then 1% uranyl acetate (Serva) in 70% ethanol for 60 min, before being embedded in Durcupan (Sigma–Aldrich) for ultrathin sectioning (60 nm thickness). Sections were collected on single slot Formvar-coated copper grids.

Images of the dentate molecular layer in tissue cultures prepared from eGFP-SP-tg mice (n = 3) were captured with a Zeiss electron microscope (Zeiss EM 900) at 20,000x magnification. Spine head area and area of corresponding spine apparatus were manually outlined with the Polygon tool and quantified using ImageSP Viewer (Version 1.2.4.22, SysProg). Statistical analysis was done with GraphPad Prism six software. Using the same materials, images of the dentate molecular layer taken at 12,000x magnification were used to determine the percentage of spine apparatuses within spines. Only dendritic profiles that could be reliably identified as spines were counted (n = 172). Spines were defined as dendritic profiles with synaptic specializations but without mitochondria. A spine apparatus was considered present, if one dense plate and two ER-cisterns were identified.

## Confocal microscopy of fixed hippocampal slices

Confocal imaging of fixed dendritic segments from identified, Alexa568-labeled dentate granule cells in the outer molecular layer (OML) of the suprapyramidal blade was done with an Olympus FV1000 microscope and a 60x oil-immersion objective (UPlanSApo, NA 1.35, Olympus) using FV10-ASW software with 5x scan zoom at a resolution of 1024 × 1024 pixels. To ensure localization in the OML, 3D image stacks of dendritic segments (0.25 µm z-axis step size) were taken at a distance of 10–50 µm from the hippocampal fissure. Crossing dendritic segments or branch points were avoided to facilitate spine attribution to a given segment.

## 2-Photon time-lapse imaging of tissue cultures

Live imaging of tdTomato-labeled granule cells with eGFP-tagged SP clusters was done using an upright 2-photon microscope (Scientifica MPSLSC-1000P, East Sussex, UK) equipped with a 40x water immersion objective (Zeiss Plan-Apochromat, NA 1.0) and a Ti-sapphire mode-locked laser (MaiTai, Spectra-Physics) tuned to a wavelength of 1000 nm to excite both eGFP and tdTomato. The membrane insert carrying the cultures was placed in a petri dish containing warm imaging buffer consisting of 129 mM NaCl, 4 mM KCl, 1 mM $MgCl_2$, 2 mM $CaCl_2$, 4.2 mM glucose, 10 mM HEPES buffer solution, 0.1 mM Trolox, 0.1 mg/ml streptomycin, 100 U/ml penicillin, and pH 7.4. 3D image stacks of dendritic segments (~50–80 µm in length) located in the middle to outer molecular layers (15–25 images per stack, 0.5 µm z-axis step size) were acquired using ScanImage 5.1 (*Pologruto et al., 2003*) with 8x digital zoom at a resolution of 512 × 512 pixels, that is 0.07 µm x

0.07 µm in the focal plane. The same dendritic segments across imaging sessions were identified using nearby landmarks and neighboring dendrites. Imaging time was minimized (only one dendritic segment per culture was imaged) to minimize the risk of phototoxic damage.

## Image processing and data analysis

The confocal as well as 2-Photon images obtained were deconvolved with Huygens Professional Version 17.10 (Scientific Volume Imaging, The Netherlands, http://svi.nl). Image processing and data analysis were then performed using Fiji version 1.52 hr (*Schindelin et al., 2012*), with spine analysis adapted from published criteria (*Holtmaat et al., 2009*). Figures were prepared using commercial software. Image brightness and contrast were adjusted.

Dendritic spines of all shapes were assessed manually on z-stacks of dendritic segments in the middle to outer molecular layers. Only protrusions emanating laterally in the x-y directions, not above or below the dendrite, and exceeding the dendrite for at least five pixels (0.35 µm for 2-Photon; 0.2 µm for confocal) were included for analysis (*Holtmaat et al., 2009*; *Vlachos et al., 2012*). We refrained from using shape-based categories in our quantitative analyses, for example stubby, thin, mushroom spines, because spines are dynamic and exhibit a continuum of sizes and shapes (*Trommald and Hulleberg, 1997*). We rather focused on the spine head, since spine head size is a critical functional parameter that can be reliably measured (*Kasai et al., 2003*; *Matsuzaki et al., 2004*; *Bosch et al., 2014*), and SP has been shown to be associated with spine head size (*Okubo-Suzuki et al., 2008*; *Vlachos et al., 2009*; *Zhang et al., 2013*) but not spine length (*Deller et al., 2003*).

For spine head size and SP cluster size measurements, the maximum cross-sectional area of the spine head or SP cluster in one of the x-y planes within the z-stack was measured. A spine was considered SP+ if the SP cluster overlapped with the spine head and/or neck in both the x-z and y-z directions when scrolling through the z-stacks. For spine stability analysis, we examined ~15 µm long subsegments exhibiting minimal dendritic distortion over the total observation period. Individual spines were re-identified at consecutive points in time based on their relative positions to nearby landmarks and neighboring spines (*Holtmaat et al., 2009*; *Vlachos et al., 2012*). The presence or absence of a particular spine at a certain time point was verified by scrolling through the z-stacks to avoid misclassification due to rotation of the dendritic segment.

Spine turnover ratio representing the fraction of all observed spines that appear and/or disappear during the total imaging period was measured (adapted from *Holtmaat et al., 2009*). Spine turnover ratio = (Ngained + Nlost) / (2 x Ntotal), where Ngained is the number of spines gained between the initial observation time point at day −1 and the final observation time point at day 12, Nlost is the number of spines lost between day −1 and day 12, and Ntotal is the total number of spines that were observed at one or more points in time. Spine formation ratio = (Ntotal − Ninitial) / (Ntotal), where Ninitial is the number of spines observed at day −1. Spine loss ratio = (Ntotal − Nfinal) / (Ntotal), where Nfinal is the number of spines observed at day 12.

## Statistical analysis

Statistical tests and n-values are indicated in figure captions. Statistical tests were chosen based on the experimental conditions, that is (1) matched-pairs, (2) two independent groups, (3) multiple matched-pairs, or (4) multiple independent groups. Thus, statistical comparisons were performed using Wilcoxon signed rank test with Pratts modification accounting for ties and zero differences in case of comparing integer values (differences between matched pairs under the null hypothesis of no difference), Mann–Whitney U-test (comparing two independent groups), Friedman test (for differences between matched pairs of multiple groups), and Kruskal-Wallis test (comparing multiple groups). Dunn's multiple comparison test was applied following one of the two latter tests in case of significant p-values. All statistical tests were performed using GraphPad Prism 6. If p-values were less than 0.05 the null hypothesis was rejected. Statistical values were expressed as mean ± standard error of mean (SEM), unless otherwise stated. *$p < 0.05$, **$p < 0.01$, ***$p < 0.001$.

## Modeling of spine survival and loss

The fractions of surviving spines observed at distinct time points were fitted to different models of spine survival and loss, respectively (see results). Curve fitting was performed with the user-defined

fitting function of GraphPad Prism 6. Decay functions were fitted to the data weighted by $1/Y^2$ using least squares fit. The resulting spine loss curves correspond to the probability that single spines experience a particular survival time. Therefore, the loss curves represent the cumulative density function of spine survival times. We derived the distribution of spine survival times underlying the spine loss curves from the inverse loss curves. Since there is no analytical solution for the inverse of the conditional two-stage decay, the inverse loss functions (i.e. the inverse cumulative density functions of spine survival times) were obtained from the spine loss models numerically as follows: According to the diverse loss models, the fractional survival was calculated for $10^7$ equally spaced time points ranging from zero to 30 times the longest time constant of the respective decay function. For each of $10^5$ equally spaced bins of fractional survival, the mean of the corresponding times (X-axis) was defined as the survival time corresponding to one minus the center of the respective fractional survival interval. The distribution of survival times displayed in the figures was derived as the histogram of these values using a bin size of 100. The median as well as the 10th, 25th, 75th, and 90th percentiles of these distributions are also displayed in the figures. Statistical comparisons of median survival times between groups were based on the medians of 10 random samples of sizes equal to the number of observed spines per group using the Mann–Whitney U-Test. Derivation of survival times was performed using LabView 2019 (National Instruments).

## Acknowledgements

We thank Charlotte Nolte-Uhl and Anke Biczysko for technical support and Dr. Stephan W Schwarzacher for critical discussion of our data. The authors (D.D.T., A.D., T.D.) dedicate this paper to the memory of the late Michael Frotscher, M.D., who was a wonderful mentor and an enthusiastic scientist and who was involved in many pioneering studies on the localization and the role of Synaptopodin in the CNS.

## Additional information

### Funding

| Funder | Grant reference number | Author |
|---|---|---|
| Deutsche Forschungsgemeinschaft | CRC 1080 | Thomas Deller |
| Deutsche Forschungsgemeinschaft | DE2741/1-1 | Domenico Del Turco |
| Deutsche Forschungsgemeinschaft | DE 551/13-1 | Thomas Deller |
| International Max Planck Research School | scholarship | Kenrick Yap |

The funders had no role in study design, data collection and interpretation, or the decision to submit the work for publication.

### Author contributions

Kenrick Yap, Data curation, Formal analysis, Funding acquisition, Validation, Investigation, Visualization, Methodology, Writing - original draft, Project administration, Writing - review and editing; Alexander Drakew, Conceptualization, Software, Formal analysis, Supervision, Validation, Visualization, Methodology, Writing - original draft, Writing - review and editing; Dinko Smilovic, Michael Rietsche, Mandy H Paul, Data curation, Formal analysis, Investigation, Visualization, Writing - review and editing; Mario Vuksic, Formal analysis, Supervision, Investigation, Writing - review and editing; Domenico Del Turco, Resources, Funding acquisition, Methodology, Writing - review and editing; Thomas Deller, Conceptualization, Resources, Formal analysis, Supervision, Funding acquisition, Visualization, Writing - original draft, Project administration, Writing - review and editing

## Author ORCIDs

Kenrick Yap (iD) http://orcid.org/0000-0003-0123-7133
Dinko Smilovic (iD) https://orcid.org/0000-0001-9348-7177
Michael Rietsche (iD) https://orcid.org/0000-0001-6265-6791
Mario Vuksic (iD) http://orcid.org/0000-0003-2196-2244
Domenico Del Turco (iD) https://orcid.org/0000-0002-9594-6186
Thomas Deller (iD) https://orcid.org/0000-0002-3931-2947

## Ethics

Animal experimentation: All animal experiments were performed in accordance with the German animal welfare law and had been declared to the Animal Welfare Officer of the Medical Faculty (Wa-2014-35). Every effort was made to minimize the distress and pain of animals.

## Decision letter and Author response

Decision letter https://doi.org/10.7554/eLife.62944.sa1
Author response https://doi.org/10.7554/eLife.62944.sa2

## Additional files

### Supplementary files

• Transparent reporting form

### Data availability

Data generated or analyzed during this study were included in the manuscript (individual data points are illustrated in the figures). Furthermore, supporting figures show single spine imaging data providing the basis for analysis.

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
