## [Decision Letter]

**Acceptance summary:**

Tracking a fluorescently-tagged SP fusion protein, the authors show that the small percent of spines that are SP positive (SP+) skew large, and are the most stable fraction of the spine population. The presence of SP in spines correlated well with prolonged survival of dendritic spines.

**Decision letter after peer review:**

[Editors’ note: the authors submitted for reconsideration following the decision after peer review. What follows is the decision letter after the first round of review.]

Thank you for submitting your work entitled "The actin-modulating protein Synaptopodin mediates long-term survival of dendritic spines" for consideration by *eLife*. Your article has been reviewed by three peer reviewers, and the evaluation has been overseen by a Reviewing Editor and a Senior Editor. The reviewers have opted to remain anonymous. Our decision has been reached after consultation between the reviewers. Based on these discussions and the individual reviews below, we regret to inform you that your work will not be considered further for publication in *eLife*.

Although the reviewers agree that this study helps us further understand the function of Synaptopodin, they find that your study does not offer sufficient novel mechanistic insights and that there are a number substantial issues that need to be addressed. If, however, you fully address the review comments and add sufficient amounts of additional data or analysis to substantially improve the manuscript, we would be willing to re-assess the study, as a new submission with a new manuscript number and submission date, although it is your choice. We realize that many labs are closed for some period and that additional experiments could be delayed for months, thus we are willing to consider a (re)submission regardless of the date.

Reviewer #1

The work by Yap et al. consists of a major effort by Deller and colleagues over the past two decades, to understand the role of synaptopodin (SP) in the formation and plasticity of dendritic spines in rodent dentate granular neurons. The factors involved in regulation of dendritic spines morphology and functions have been the focus of numerous studies in the past two decades, employing a variety of tools, from molecular to high power imaging methodologies, both in-vivo and in-vitro. Formation, density, size, shape and longevity of spines have been associated with functional plasticity of neurons and networks, learning and memory. Conflicting results in this field probably reflect differences in brain regions, preparation, age, optical resolution and past history of the neuron.

An enigmatic organelle, the spine apparatus, regained intensive interest, since it was associated with SP, which can be visualized in live tissue. The group of Deller made already significant contributions in the field to demonstrate that SP is related to spine plasticity. The present work, using normal and SP-KO mice, clarifies how SP may be related to the morphological properties of the spines. Time-lapse two photon microscopy of organotypic slices, Yap et al. were able to follow individually identified dendritic spines over two weeks in culture. Carefully analyzing changes in the morphology and SP in thousands of spines, they present some very appealing observations on the role of SP in the development, and more importantly, in pruning of spines of granular neurons. Briefly, they show that SP-containing spines (SP+) live longer that SP(-) spines, that they are more likely to reside in larger spines, consistent with the idea that large spines are loci of "memory". They suggest that pruning of SP+ spines is a two stage process, involving first the disappearance of SP, followed by shrinkage of the spine. SP(-) spines shrink in a mono-exponential time constant. Thus, they propose that SP is a major regulator of spine longevity (=memory).

The study is extensive, carefully conducted and clearly presented, and should be accepted for publication; provide the authors address the following issues:

In earlier studies by Deller et al., no difference was found in distribution of spine size between SP− and SP+ spines. Assuming that SP+ spines decay much slower than SP− spines, one would expect an increase in spine size/density in SP+ cells, but they did not find this. Any explanation?

In earlier studies, SP− slices produce or do not produce as large LTP as SP+ slices. This may be an age-dependent effect. In the present in-vivo study, they do not report any age factor. Did they look for this? In any case, they cannot say categorically that SP− animals are less able to learn.

In the current study, they do not separate between mature spines, filopodia, stubby or short spines. Could they comment on the possible SP dependent conversion of stubby spines to mature spines?

Neither do they comment on spine density in the different conditions. Is it possible that dendritic spine density changes contribute to the different proportion of spine shape across age?

Reviewer #2:

The paper by Yap et al. is generally well written. The message is clear and experiments are straightforward and well conducted. I do not have any major concern on the manuscript. One thing the author should add, in either Discussion or Introduction, is the description of protein domain structure and possible interaction partners, in the context of observed effect on spine turnover. I realize that it is not the specialty of the authors but still it is important to give some idea. Interestingly, synaptopodin accumulate at the bottom of spine head. In view of its function, possible mechanism to stabilize spine may be discussed. This reviewer knows of only one other example which shows such distribution – cofilin (Bosch, 2014). Another thing the authors can add to Discussion or Introduction is about regulation of synaptopodin by neuronal activity. Although it is beyond the current study, actin is dynamically regulated during synaptic plasticity. Therefore, it is important to understand how synaptopodin is regulated and what the mechanism might be.

Having said this, the Discussion is lengthy. The authors do not need to repeat the results.

Reviewer #3:

The manuscript continues the studies of these authors on the actin-stabilizer synaptopodin (SP). This manuscript describes a role for SP in the stability of dendritic spines in the OML of dentate granule cells in organotypic slice cultures. SP+ spines were assessed in Thy1-EGFP-SP mice on a SP KO background The authors have previously showed that SP is a component of the spine apparatus and SP KO mice have deficits in synaptic plasticity. The main new finding in this manuscript is that larger spines contained SP and correlated with longer survival than SP− spines as measured with two-photon microscopy. The experiments appear carefully executed and add to descriptions of synaptopodin in spines, but represent detailed, but modest, extension of prior data. The observations regarding loss of spines based on following spines over time is interesting, but does not provide particular mechanistic insight as the paper is entirely SP-centric without consideration for other molecular interactions in spines. There are also a few issues of interpretation as below.

1) SP was expressed in SP KO mice and labeled approximately 14% of the total spines in tissue sections, but without changing the average size, perhaps not totally surprising (although the authors state it is surprising), given that only a small fraction of the spines were labeled. Given the small fraction of SP-labeled spines, it seems the authors conclusion that SP is not "…a major regulator of spine head size…" is not proven by this data. The same concern applies to the organotypic slice date where only 8% of spines were SP+.

2) The authors claim that SP was not overexpressed based on mRNA of the culture, but presumably only a fraction of cells were labeled. Thus measurements of virally-tranfected cells would be more convincing.

3) Portions of the Discussion reiterate the results and could be substantially shortened.

[Editors’ note: further revisions were suggested prior to acceptance, as described below.]

Thank you for resubmitting your work entitled "The actin-modulating protein Synaptopodin mediates long-term survival of dendritic spines" for further consideration by *eLife*. Your revised article has been evaluated by Gary Westbrook (Senior Editor) and three reviewers.

The manuscript has been improved but there are a few remaining issues that need to be addressed before acceptance. These include modifications in the text to address the concerns raised by reviewer 3 and to a lesser extent reviewer 2. These are important revisions to place your results in the context of the field. The editor will evaluate your response and your manuscript will not be sent back to the reviewers.

Reviewer #1:

The authors adequately revised the paper according to my suggestions. It should be ready for publication.

Reviewer #2:

Overall this is a well done revision that in my mind addresses all of the prior reviewer concerns. The finding that Synaptopodin influences spine stability is well supported by comparisons of SP+ versus SP− spine lifetimes as well as comparisons between WT and KO mice. Overall these findings are of interest to the field, but it is slightly disappointing there is no mechanistic insight accompanying the observations.

Reviewer #3:

In this manuscript, the authors expand on their previous studies of the protein synaptopodin (SP), an essential component of the spine apparatus, an organelle that modulates second messenger signaling in dendritic spines and implicated in F-actin. Following their findings on a short term role for SP in plasticity-induced spine head expansion, here the authors examine a long term role for in spine stability by time lapse imaging of individual spines on dentate granule cells in organotypic and acute mouse hippocampal slices.

Tracking a fluorescently-tagged SP fusion protein, they show that the small percent of spines that are SP positive (SP+) skew large, and are the most stable fraction of the spine population. The presence of SP in spines correlates well with prolonged survival, irrespective of spine size, and even after denervation. SP+ spines that underwent pruning first lost synaptopodin, following a two-stage exponential decay kinetics, vs SP− spines that follow a single exponential decay. Based on a mathematical analysis, they present a model compatible with the pruning of SP+ spines via a two-stage process in which they first lose their synaptopodin, converting to SP− spines that are then pruned with the same time constant as spines that start SP-. The data summarizing these findings is clearly presented, and represents meticulously done and well-controlled experiments.

There are only two points that diminish my enthusiasm.

1) First, aside from Figure 6, the findings presented are all correlative, and the model is based on this correlative data. Figure 6 presents a spine size/stability analysis in SP KO neurons, showing that spines in the KO essentially behave as SP− spines in the WT. This figure is the only one that tests the requirement for SP for spine stability, and on its own is not definitive, especially given the homeostatic compensation that partially counteracts the constitutive SP loss. Considering this, some of the statements they make are a bit too strong. For example, the heading for Figure 5 "Presence of SP in spines determines their long-term survival". The data shown in this figure is that presence of SP is correlated with long-term survival, there is no test of sufficiency that would justify use of the word determines. The heading is similarly misleading: “Synaptopodin increases survival of small, medium and large spines”. The authors should carefully peruse the manuscript and correct all such statements that claim more than a correlation unless it is explicitly tested (such as in Figure 6).

2) A second disappointment is that the entire paper is oddly removed from a deep literature related to other molecular determinants of spine stability. In particular, the relationship between PSD95 presence and spine size and stability has been comprehensively analyzed, and PSD95 presence is considered one of the best predictors of spine stability. While I would not ask for any mechanistic analysis connecting SP to other molecules previously shown to determine spine stability, I am surprised that neither the Introduction or Discussion even mention the molecular precedents for their finding.

---

## [Author Response]

[Editors’ note: the authors resubmitted a revised version of the paper for consideration. What follows is the authors’ response to the first round of review.]

Reviewer #1[…]The study is extensive, carefully conducted and clearly presented, and should be accepted for publication; provide the authors address the following issues:In earlier studies by Deller et al., no difference was found in distribution of spine size between SP− and SP+ spines. Assuming that SP+ spines decay much slower than SP− spines, one would expect an increase in spine size/density in SP+ cells, but they did not find this. Any explanation?

The reviewer refers to our publication in 2003 (Deller et al., 2003). In this publication we used Golgi-staining to label single pyramidal cells in CA1, CA3, cortex and striatum in wildtype and SP-deficient mice. We tested for differences in spine density and spine length (investigator blind to genotype), which we did not observe. Later, three other labs also investigated spine densities under conditions of SP-overexpression (Okubo-Suzuki et al., 2008; Inokuchi lab; Figure 4), SP-knock-down (Vlachos et al., 2009; Segal lab) and SPdepletion in spines using dominant-negative Myosin V mutations (Konietzny et al., 2019; Mikhaylova lab) and confirmed that SP does not affect spine densities.

The most likely explanation for the fact that no differences in spine densities are observed in mice with altered SP-levels appears to be a homeostatic mechanism that counteracts changes in SP-mediated spine stability: Decreases in spine stability are compensated for by an increase in spine formation and turnover rate (seen in Figure 2—figure supplement 2). This explains why the loss of ~10% stable spines in SP-deficient mice does not cause a decrease in spine density (which was our initial prediction, too) or why an overexpression of SP does not cause an increase in average spine density (Okubo-Suzuki et al., 2008; our study). This is now mentioned in the text.

Regarding spine head size, Okubo-Suzuki et al., 2008, were the first to show a positive correlation between SP, F-actin content and spine volume (DsRed2-transduced cells). Shortly thereafter, Vlachos et al., 2009, reported a similar relationship between spine head size and SP. Both studies used primary neuronal cultures. Our study demonstrates the validity of these observations in an organotypic slice culture environment and links the presence of SP to spine stability. Thus, our present study adds considerably to our knowledge in this field.

In earlier studies, SP− slices produce or do not produce as large LTP as SP+ slices. This may be an age-dependent effect. In the present in-vivo study, they do not report any age factor. Did they look for this? In any case, they cannot say categorically that SP− animals are less able to learn.

We thank the reviewer for pointing this out. Age-dependence plays a role because SP is gradually expressed in the hippocampus during the postnatal period, starting around the first week postnatally (Czarnecki et al., 2005). It is strongly expressed during adult life (Czarnecki et al., 2005; Deller et al., 2000) and is downregulated with aging (Sidhu et al., 2016). We have now mentioned this age-dependence in the revised manuscript.

As far as LTP-impairment is concerned, an LTP-impairment has been shown in juvenile (15-21 day old; Zhang et al., 2013) as well as in adult (Deller et al., 2003; Jedlicka et al., 2009) mice. In our first study on the SP-KO mouse (Deller et al., 2003), we also investigated spatial learning and found that adult SP-KO mice perform less well in the radial arm maze compared to wildtype mice, suggesting that lack of SP results in a learning/memory phenotype. The reviewer is right, however, that these findings have not been confirmed at other ages and we have now toned down our statement and have made it clear in the revised manuscript that spatial learning deficits were reported previously in adult SP-KO mice.

In the current study, they do not separate between mature spines, filopodia, stubby or short spines. Could they comment on the possible SP dependent conversion of stubby spines to mature spines?

We thank the reviewer for allowing us to comment on this issue. We have not used the classical classification of spines in our work, i.e. the classification of spines into four categories: thin, stubby, mushroom and (sometimes) filopodia. This classification is based on fixed tissues (“snapshots capturing states of spines”) and the definition of spine classes varies considerably between labs.

In fact, detailed analyses of a large number of spines revealed that spines come in a continuum of sizes and shapes (e.g. Trommald and Hulleberg, 1997; J Comp. Neurol., 377:15-28). The four classical categories lose their meaning in a dynamic environment, in which spines constantly remodel their geometry (e.g., Matus, 1999, Curr. Opin. Neurobiol. 9:561-5), and in which – as the reviewer points out – spines shift from one category into another. Indeed, as mentioned by the reviewer, stubby spines can transform into mushroom spines with the insertion of SP. This was shown by us in an earlier publication (Vlachos et al., 2009). In Figure 2D of this earlier publication (Vlachos et al., 2009), we show in a sequence of images the transformation of a SP− stubby spine into a SP+ mushroom spine. We found similar examples in our material. In the revised manuscript we have now explained our reasoning for choosing spine head size rather than spine categories as readout parameter.

Neither do they comment on spine density in the different conditions. Is it possible that dendritic spine density changes contribute to the different proportion of spine shape across age?

As mentioned in our response to the second point raised by the reviewer, we concur that aging and maturation play an important role and we have now mentioned this in our revised manuscript.

The suggestion of the reviewer is very interesting, since an age-associated reduction in spine stability could contribute to cognitive impairment or learning and memory deficits. Indeed, it has recently been shown that SP expression levels decrease with age (Sidhu et al., 2016) and that SP levels are strongly reduced under conditions of cognitive impairment or Alzheimer´s disease (Reddy et al., 2005; Counts et al., 2014; Wingo et al., 2019). These changes in SP correlate with a reduced spine stability in aged animals (Voglewede et al., 2019) and in AD model mice (Spires-Jones et al., 2007), raising the possibility that these two phenomena are linked. We have now mentioned this intriguing possibility in the revised manuscript and thank the reviewer for this suggestion.

Reviewer #2:The paper by Yap et al. is generally well written. The message is clear and experiments are straightforward and well conducted. I do not have any major concern on the manuscript. One thing the author should add, in either Discussion or Introduction, is the description of protein domain structure and possible interaction partners, in the context of observed effect on spine turnover.

We thank the reviewer for this suggestion and have included pertinent information on the published sequence, the structure, organ-specific SP-isoforms and potential binding partners in the revised manuscript (Mundel et al., 1997; Asanuma et al., 2005; 2006; Faul et al., 2007). SP lacks a clear domain organization and contains several predicted disordered regions (Asanuma et al., 2005; Chalovich and Schroeter, 2010; Konietzny et al., 2019).

Several groups have identified potential interaction partners of SP using 2-hybrid screens (Kremerskothen et al., 2005; Asanuma et al., 2006) and mass spectrometric analysis of a pulldown fraction from hippocampus (Konietzny et al., 2019). Brain SP binds actin (Mundel et al., 1997; Asanuma et al., 2006; Okubo-Suzuki et al., 2008) and α actinin2 (Asanuma et al., 2005; Kremerskothen et al., 2005) in all cellular compartments in which SP is found (Paul et al., 2020). Other potential interaction partners that could affect spine stability are Cdc42 and RhoA (Asanuma et al., 2005; Kremerskothen et al., 2005; Asanuma et al., 2006; Yanagida-Asanuma et al., 2007; Faul et al., 2007; Konietzny et al., 2019). Recently, myosin V has been identified as an interaction partner by Konietzny et al., 2019, together with one of us (Alexander Drakew). In this recent study several novel interaction partners were also identified using mass spectrometric analysis of a pulldown fraction from rat hippocampus and a network analysis was published (Figure 1B in Konietzny et al., 2019). We refer the reviewer to this figure for additional details on potential interaction partners of SP.

I realize that it is not the specialty of the authors but still it is important to give some idea. Interestingly, synaptopodin accumulate at the bottom of spine head. In view of its function, possible mechanism to stabilize spine may be discussed. This reviewer knows of only one other example which shows such distribution – cofilin (Bosch, 2014).

We fully concur with the reviewer and have taken up this suggestion. We have now added a paragraph to the Discussion, in which we discuss possible mechanisms and interaction partners that could allow SP to stabilize spines. This paragraph now reads:

“How could SP exert its stabilizing effects on spines? The preferential localization of SP clusters in the lower part of the spine head argues for an association of SP with the stable pool of actin, which forms the central and less dynamic core of spines (Bramham et al., 2008; Bosch et al., 2014; Colgan and Yasuda, 2014; Spence and Soderling, 2015). […] In sum, SP could influence the stability of actin pools and/or actin treadmilling in spines via several mutually non-exclusive direct and indirect mechanism.”

Another thing the authors can add to Discussion or Introduction is about regulation of synaptopodin by neuronal activity. Although it is beyond the current study, actin is dynamically regulated during synaptic plasticity. Therefore, it is important to understand how synaptipodin is regulated and what the mechanism might be.

We thank the reviewer for making this point. Activity-dependent regulation of SP has been shown by others, in particular the group of Inokuchi. They showed that under conditions of synaptic strengthening SP is upregulated (Yamazaki et al., 2001) and is also sorted into the same layer of the hippocampus as actin (Fukazawa et al., 2003). In a slightly later paper, the same group showed that activation of NMDA-R results in SP recruitment into spines (Okubo-Suzuki et al., 2008). Within spines SP stabilized enlarged spine heads. Finally, using a chemical LTP approach, the group of Menahem Segal in collaboration with our group (Vlachos et al., 2009) showed that SP is recruited into spines under these conditions. The number of SP-positive spines increased significantly.

In sum, these data show that SP is regulated by activity. It locally accumulates below activated synapses and is recruited into synapses upon activity-induced spine head expansion. We have now discussed this in the revised manuscript on Hebbian plasticity (activity regulation) and the revised paragraph now reads:

“SP plays an important role in Hebbian- (Yamazaki et al., 2001; Deller et al., 2003; Okubo-Suzuki et al., 2008; Holbro et al., 2009; Jedlicka et al., 2009; Vlachos et al., 2009; Zhang et al., 2013; Jedlicka and Deller, 2017) and homeostatic (Vlachos et al., 2013) forms of synaptic plasticity. […] Since SP+ spines are large and strong spines exhibiting a high density of AMPA-R (Vlachos et al., 2009), loss of SP from only this fraction of spines suffices to impair Hebbian plasticity (Deller et al., 2003; Jedlicka et al., 2009; Vlachos et al., 2009; Zhang et al., 2013; Grigoryan and Segal, 2015), underlining its role in Hebbian forms of synaptic strengthening.”

Having said this, the Discussion is lengthy. The authors do not need to repeat the results.

We have followed the reviewer´s recommendation and have included the above paragraphs and have removed others.

Reviewer #3:The manuscript continues the studies of these authors on the actin-stabilizer synaptopodin (SP). This manuscript describes a role for SP in the stability of dendritic spines in the OML of dentate granule cells in organotypic slice cultures. SP+ spines were assessed in Thy1-EGFP-SP mice on a SP KO background The authors have previously showed that SP is a component of the spine apparatus and SP KO mice have deficits in synaptic plasticity. The main new finding in this manuscript is that larger spines contained SP and correlated with longer survival than SP− spines as measured with two-photon microscopy. The experiments appear carefully executed and add to descriptions of synaptopodin in spines, but represent detailed, but modest, extension of prior data. The observations regarding loss of spines based on following spines over time is interesting, but does not provide particular mechanistic insight as the paper is entirely SP-centric without consideration for other molecular interactions in spines. There are also a few issues of interpretation as below.

We agree with the reviewer that this work is highly focused on SP. We do, in fact, regard this as a strong point of the paper, because we have finally uncovered what SP does with regard to spine dynamics. After the group of Inokuchi showed that SP does not cause spine head expansion but stabilizes expanded spine heads (Okubo-Suzuki et al., 2008), a role for SP in the regulation of spine stability was one of the discussed functions.

To resolve this open question, we performed an in-depth analysis of the biological effects of SP on spines in an organotypic environment. Mechanistic interactions, e.g. interaction of SP with other actin-binding proteins can be much better investigated in more simple systems, e.g. dissociated cultures, which have their own limitations. In fact, some of us recently contributed to a report on molecular interaction partners of SP using such an approach (Konietzny et al., 2019).

1) SP was expressed in SP KO mice and labeled approximately 14% of the total spines in tissue sections, but without changing the average size, perhaps not totally surprising (although the authors state it is surprising), given that only a small fraction of the spines were labeled. Given the small fraction of SP-labeled spines, it seems the authors conclusion that SP is not "…a major regulator of spine head size…" is not proven by this data. The same concern applies to the organotypic slice date where only 8% of spines were SP+.

We concur with the reviewer that non-significant results need to be carefully interpreted. However, in light of the fact that the biggest spines of a neuron are SP+ spines, it was unexpected for us that absence of SP from granule cells neither affected average spine head size nor cumulative spine distribution (Figure 1C, D). Even if only 13.5% of spines contain SP, we had anticipated that such an effect should be visible, because SP+ spines are three times the size of SP− spines (Figure 1G; ~0.1 Mean spine head size of SP− (~0.133 μm^2^) and SP+ (~0.328 μm^2^) spines. ***p < 0.0001, Mann–Whitney U-test. SP+ spines n = 200; SP− spines n = 1497.).

Our data suggest that one reason for this negative finding, i.e. non-significant difference in average spine head size between KO and WT mice, is a compensation mechanism in the KO mouse that could only be detected using time-lapse imaging: SP-deficient granule cells show an increased spine turnover (Figure 6—figure supplement 1). Unstable spines disappear but are rapidly replaced by new spines, thus compensating not only for spine density but also for spine head size (see our response to reviewer 1). These read-out parameters may be homeostatically regulated, e.g. by the global firing rate of a neuron (Turrigiano et al., 2008; 2011).

Based on our observation, that the distribution between SP-KO mice and SP-WT mice is not significantly different, we do not have any data which allow us to state the opposite, i.e. that SP is a major regulator of spine head size. We, therefore, in line with our data carefully stated that SP is “not a *major* regulator of spine head size”, which is what our data shows. This statement does not exclude a regulatory role of SP for spine head size that remained undetected in this context.

To address the concerns of the reviewer, we have been even more conservative in our wording. We changed the text and toned down our statements. Specifically, the text passage at the end of the first Results paragraph now reads:

“We conclude from these observations, that SP is indeed tightly correlated with spine size. However, lack of SP did not affect average spine head size, suggesting that SP is either not a major regulator of spine head size or that its loss is compensated for in the SP-deficient mouse.”

2) The authors claim that SP was not overexpressed based on mRNA of the culture, but presumably only a fraction of cells were labeled. Thus measurements of virally-tranfected cells would be more convincing.

We thank the reviewer for this comment. We have performed additional experiments to address these concerns. First, we performed fluorescence in situ hybridization (FISH) for SP-mRNA. Since the cultures lack endogenous SP, this revealed all cells expressing the transgene. In the dentate gyrus granule cell layer, essentially every GC expressed the transgene. Cultures from litter mate controls lacking the transgene (which are, in essence, KO) did not show a SP signal. We have now mentioned this in the revised manuscript and have added Figure 2—figure supplement 1.

Secondly, we micro-dissected areas of the granule cell layer injected with the tdTomato virus and compared the level of SP-mRNA in virus-injected tissues of eGFP-SP-tg and control tissues. Again, levels of SPmRNA were not significantly different, supporting our earlier findings and the results of the FISH. We have now mentioned this in the revised manuscript and have added the data to Figure 2—figure supplement 1.

Finally, we would like to add that we investigated the stability of single SP+ spines, not cells. SP is transported to spines where the protein exerts its effects locally. Thus, even if there are differences in expression in SP-mRNA between single cells or even in SP protein between spines, this would not invalidate the major finding of our study, i.e. an effect of SP on long-term spine stability, which is seen in SP+ but not in SP− spines or in spines from KO mice.

3) Portions of the Discussion reiterate the results and could be substantially shortened.

We have shortened the Discussion and have focused on our main findings and the additional topics suggested by reviewer 2. See also our response to reviewer 2.

[Editors’ note: what follows is the authors’ response to the second round of review.]

The manuscript has been improved but there are a few remaining issues that need to be addressed before acceptance. These include modifications in the text to address the concerns raised by reviewer 3 and to a lesser extent reviewer 2. These are important revisions to place your results in the context of the field. The editor will evaluate your response and your manuscript will not be sent back to the reviewers.Reviewer #3:[…]The data summarizing these findings is clearly presented, and represents meticulously done and well-controlled experiments.

We thank the reviewer for the positive feed-back and for the time spent reviewing our manuscript.

There are only two points that diminish my enthusiasm.1) First, aside from Figure 6, the findings presented are all correlative, and the model is based on this correlative data. Figure 6 presents a spine size/stability analysis in SP KO neurons, showing that spines in the KO essentially behave as SP− spines in the WT. This figure is the only one that tests the requirement for SP for spine stability, and on its own is not definitive, especially given the homeostatic compensation that partially counteracts the constitutive SP loss. Considering this, some of the statements they make are a bit too strong. For example, the heading for Figure 5 "Presence of SP in spines determines their long-term survival". The data shown in this figure is that presence of SP is correlated with long-term survival, there is no test of sufficiency that would justify use of the word determines. The heading is similarly misleading: “Synaptopodin increases survival of small, medium and large spines”. The authors should carefully peruse the manuscript and correct all such statements that claim more than a correlation unless it is explicitly tested (such as in Figure 6).

We have gone through our manuscript and have performed the suggested changes. Tracking mode changes (or text passages highlighted in yellow) indicate where we have changed the wording. Specifically, the heading for Figure 5 now reads: "Presence of Synaptopodin correlates with their long-term survival". The heading now reads: "Synaptopodin is correlated with the long-term survival of spines irrespective of spine head size".

2) A second disappointment is that the entire paper is oddly removed from a deep literature related to other molecular determinants of spine stability. In particular, the relationship between PSD95 presence and spine size and stability has been comprehensively analyzed, and PSD95 presence is considered one of the best predictors of spine stability. While I would not ask for any mechanistic analysis connecting SP to other molecules previously shown to determine spine stability, I am surprised that neither the Introduction or Discussion even mention the molecular precedents for their finding.

We agree with the reviewer that PSD-95 is an important predictor of spine stability. However, PSD-95 is a postsynaptic density (PSD) protein, which Synaptopodin is not. In fact, Synaptopodin has even been used by others as a marker protein for molecules located *outside* the PSD but still within the spine compartment (Hafner et al., 2019; see Supplementary Figure 6A in Hafner et al., 2009).

A number of molecules located at the synapse, e.g. several PSD molecules but also cell adhesion and/or signaling molecules spanning the synaptic cleft have been shown to stabilize excitatory synapses and spines (e.g., El-Husseini et al., 2000; Ehrlich et al., 2007; De Roo et al., 2008; Yoshihara et al., 2009; Cane et al., 2014; Meyer et al., 2014). It is highly likely that these molecules primarily act and interact at the synapse itself, whereas Synaptopodin appears to be more associated with the central and less dynamic actin core of spines (Bramham et al., 2008; Bosch et al., 2014; Colgan and Yasuda, 2014; Spence and Soderling, 2015). Indeed, our denervation experiment shows that the stabilizing effect of SP on spines is still seen even after the presynapse has been removed. Thus, Synaptopodin and PSD/synaptic molecules may differ in the mechanisms used for spine stabilization, which explains, why we have not focused in our discussion on these molecules.

However, we agree that from a systems perspective, molecules enriched in dendritic spines are likely to interact in some way and that PSD molecules could be linked to Synaptopodin, e.g. via the actin cytoskeleton. In turn, Synaptopodin could be part of their downstream machinery used for spine stabilization. To take these aspects and the suggestions of the reviewer into account, we have revised the text of our manuscript.

The text now reads:

Introduction:

“The function of SP in spines is not limited to the formation of spine apparatus organelles. SP also affects the actin spinoskeleton either directly by stabilizing F-actin (Mundel et al., 1997; Okubo-Suzuki et al., 2008) or indirectly via binding to α-actinin-2, Cdc42, RhoA, or myosin V (Asanuma et al., 2005; Kremerskothen et al., 2005; Asanuma et al., 2006; Faul et al., 2007; Yanagida-Asanuma et al., 2007; Jedlicka and Deller, 2017; Konietzny et al., 2019). It is likely that SP, via its connection to the actin cytoskeleton, is also connected to the post-synaptic density (PSD), which plays an important role in synaptic stabilization (El-Husseini et al., 2000; Ehrlich et al., 2007; Yoshihara et al., 2009; Meyer et al., 2014). Accordingly, SP has been suggested to influence the geometry and stability of spines (Deller et al., 2000).”

Discussion:

“How could SP exert its stabilizing effects on spines? In comparison to other wellestablished stabilizers of spines, such as PSD-95 (El-Husseini et al., 2000; Cane et al., 2014), which primarily act at the synapse (Ehrlich et al., 2007; De Roo et al., 2008; Yoshihara et al., 2009; Meyer et al., 2014), the preferential localization of SP clusters in the lower part of the spine head argues for an association of SP with the stable pool of actin, which forms the central and less dynamic core of spines (Bramham et al., 2008; Bosch et al., 2014; Colgan and Yasuda, 2014; Spence and Soderling, 2015).”